**JGP** Journal of General Physiology

Contractile Function

# Glycerol storage increases passive stiffness of muscle fibers through effects on titin extensibility

Seong-Won Han[1], Justin Kolb[1], Gerrie P. Farman[1], Jochen Gohlke[1], and Henk L. Granzier[1]

**To study the physiological and pathological mechanisms of muscle, it is crucial to store muscle samples in ways that preserve their properties. Glycerol is commonly used for storage, as it stabilizes muscle proteins, slows enzymatic activity, and minimizes degradation. However, previous studies validating glycerol storage have not examined its effects on passive properties. In this study, mouse extensor digitorum longus (EDL) muscles were stored in 50% glycerol in relaxing solution with protease inhibitors for various durations, then rehydrated in physiological solutions to assess mechanical properties. Active properties remained unchanged, but passive stress was sensitive to glycerol storage, showing a 56.5 ± 13.6% increase after 4 days, and this effect was permanent. The increase was most pronounced at sarcomere lengths, where titin's PEVK segment extension dominates. Using gelsolin, we determined whether the passive stress increase requires the thin filament, which is known to interact with titin's PEVK region. Both glycerol-stored fibers with and without thin filament extraction exhibited increased passive stress, suggesting that the underlying mechanism is intrinsic to titin. Finally, fibers treated with methylglyoxal, a reactive carbonyl and glycating agent that forms cross-links on lysine residues, showed a significant increase in passive stress in fibers stored in relaxing solution but not in glycerol. Thus, glycerol storage elevates passive stress in a titin-specific manner, likely involving lysine residues in the PEVK. Therefore, glycerol storage should be avoided when assessing passive stiffness. We further showed that, for long-term preservation, storage of rapidly frozen muscle at −80°C is a viable option.**

## Introduction

To study muscle physiology and the mechanisms of muscle diseases, studies often investigate the contractile properties of single muscle fibers (Ottenheijm et al., 2009; Claflin et al., 2016; Lim et al., 2019; Gohlke et al., 2024). These fibers can be isolated from skeletal muscles obtained from animal models or biopsies from patients. Since only a limited number of fibers can be analyzed in a single day, muscles or biopsies are typically stored, allowing fibers to be studied over several days to weeks (Claflin et al., 2016). This requires effective storage methods that maintain the contractile properties of the samples. Storing demembranated ("skinned") muscle in glycerol is a widely used technique in physiological studies to preserve muscle samples for extended periods (weeks to months) (Claflin et al., 2016). Glycerol acts as a cryoprotectant, preventing ice crystal formation that could otherwise damage cell structures and proteins. During glycerol storage, skinned muscle fibers are typically immersed in a solution with a high glycerol concentration (around 50%) at a low temperature, usually −20°C. This method stabilizes muscle proteins and slows enzymatic activity, reducing protein degradation.

Glycerol is generally considered nonreactive with proteins. Its molecular structure ($C_3H_8O_3$) contains three hydroxyl (OH) groups that allow glycerol to form hydrogen bonds with water molecules and protein surfaces, stabilizing proteins and reducing their tendency to denature or aggregate. This interaction usually does not involve covalent bonding with proteins, thus preserving protein function. Consequently, it is commonly assumed that skinned muscle fibers retain normal contractile function under glycerol storage conditions. However, this assumption is based on limited studies from decades ago (Szent-Gyorgyi, 1949, 1950; Benson et al., 1958), and that did not address whether glycerol storage might alter passive stiffness.

Incidental observations that we made in our laboratory revealed unexpectedly high passive stiffness in fibers stored in glycerol (Kolb et al., unpublished data). Passive stiffness in skinned muscle fibers primarily originates from the elastic I-band region of titin, which includes two main extensible elements, the tandem Ig segment (serially linked Ig-like domains) and the PEVK region (Linke et al., 1996; Trombitas et al., 1998a,

[1]Department of Cellular and Molecular Medicine, Molecular Cardiovascular Research Program, University of Arizona, Tucson, AZ, USA.

Correspondence to Henk L. Granzier: granzier@arizona.edu

This work is part of a special issue on Myofilament Structure and Function.

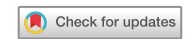

1998b; Bang et al., 2001). The PEVK region is particularly significant in skeletal muscle, as its contribution to titin's extensibility dominates across much of the physiological sarcomere length (SL) range (Trombitas et al., 1998b). This region is rich in proline (P), glutamate (E), valine (V), and lysine (K) residues (Bang et al., 2001), with lysine being of interest in this study, since lysines can react with glycating agent to form cross-links. We speculated that glycerol might react with lysine residues, potentially forming nonenzymatic covalent bonds with lysine amino groups. Additionally, cross-linking lysine residues that are distant from each other in the amino acid sequence may be feasible because the PEVK element is largely a random coil structure.

To investigate the effect of glycerol storage on passive stiffness, we examined single fibers dissected from the mouse extensor digitorum longus (EDL) muscle, also including measurements of active stress and crossbridge kinetics for completeness. As shown in detail in the results below, glycerol storage does not affect active stress or crossbridge kinetics but does increase passive stress. To determine whether this increase results from an intrinsic effect within titin or involves titin–thin filament cross-linking, fibers were treated with gelsolin to remove actin and were then stored in glycerol (Granzier and Wang, 1993b). Glycerol storage was found to increase passive stress in gelsolin-treated fibers. The SL dependence of the passive stress increases following gelsolin storage, indicating that the effect is most prominent in the length range where PEVK extension is known to be dominant. We hypothesized that glycerol might react with lysine residues in the PEVK region (e.g., 286 lysine residues in mouse skeletal muscle PEVK [Brynnel et al., 2018]). To test this, we performed experiments with methylglyoxal, a reactive carbonyl and glycating agent known to form cross-links on lysine through the Maillard reaction (Nagaraj et al., 1996). Methylglyoxal significantly increased the passive stiffness of fibers stored in relaxing solution but had no effect on fibers previously stored in glycerol.

## Materials and methods

### Animals and experimental procedures

All experiments in this study were conducted using 4-mo-old wild-type male mice (C57BL/6). Mice were anesthetized with isoflurane and euthanized via cervical dislocation. The EDL muscles were dissected from both hindlimbs and stored overnight in skinning solution (composition detailed below). The following day, the EDL muscles were thoroughly rinsed in relaxing solution (composition detailed below) and subsequently transferred to either relaxing solution or a 50% glycerol/relaxing solution mix. Samples were stored at 4°C (with replacing the relaxing solution every 24 h) or –20°C, respectively. All procedures were approved by the Institutional Animal Care and Use Committee at the University of Arizona.

### Fiber mechanics

To evaluate the effect of glycerol storage over time, the mechanical properties of single muscle fibers were assessed following storage durations ranging from 1 to 14 days ($n$ = 8–10 for

each storage solution per time point; biological variables). On the day of the experiment, individual muscle fibers were transferred to fresh relaxing solution, and their ends were attached to T-clips for mounting onto a force transducer (403A; Aurora Scientific) and a high-speed length controller (802D-120–322-TJ; Aurora Scientific). SL was measured using a high-speed VSL camera integrated with ASI 900B software (Aurora Scientific). The fiber's cross-sectional area (CSA) was determined at a SL of 2.4 µm by measuring width and height (via a right-angle prism) at three different locations along its length. The obtained values were averaged, and these values were used to calculate CSA by assuming the CSA shape to be elliptical.

### Passive force measurements

Passive force, including peak, elastic, and viscous components, was measured at SLs between 2.4 and 3.6 µm. Initially, the fiber was set to slack length to define zero force and then adjusted to 2.4 µm to determine the CSA. The fiber was passively stretched by 5% at a rate of 1%/s, followed by a 20-s hold period. This stretch protocol was repeated 10 times, incrementally extending the SL to 3.6 µm before returning to 2.4 µm (detailed protocol illustrated in Fig. 1 A).

### Active force measurements

After a 5-min rest at slack length, the fiber was stretched to a SL of 2.6 µm and activated (see Solution section). The fiber was held in the activated state until steady-state active force was achieved. Deactivation was then induced using relaxing solution (protocol outlined in Fig. 3 A). Both passive and active forces were normalized to the fiber's CSA to calculate stress (expressed in mN/mm²).

### Crossbridge kinetics

Crossbridge kinetics were quantified by measuring the rate of tension redevelopment ($K_{tr}$; Fig. 2 A). The muscle fiber was activated at a SL of 2.4 µm until steady-state active force was reached. At this point, the fiber was rapidly shortened by 20%, held for 0.02 s, and then quickly stretched back to 2.4 µm. $K_{tr}$ was determined by fitting a single-exponential regression to the tension redevelopment curve.

### Cardiomyocytes mechanics

To examine the effect of glycerol storage on a muscle type different from skeletal muscle, passive stiffness of skinned cardiomyocytes was studied. After harvesting mouse hearts ($n$ = 4), the left ventricle was frozen in liquid nitrogen, stored in –80°C for 1 mo, and then thawed and permeabilized overnight in skinning solution (see the Solutions section). On the following day, the left ventricle was cut into smaller pieces, placed in either relaxing solution ($n$ = 15) or glycerol/relaxing mixture solution ($n$ = 33), and stored at 4°C or –20°C, respectively. For the samples stored in glycerol, experiments were conducted after 3–7 days of storage. Passive force measurements were performed as described above, except the initial SL of cardiomyocytes was set at 1.9 µm and cells were stretched stepwise to a maximal SL of 2.45 µm.

### Calcium sensitivity

Calcium sensitivity of skeletal muscle fibers was examined by measuring active force at various calcium concentrations

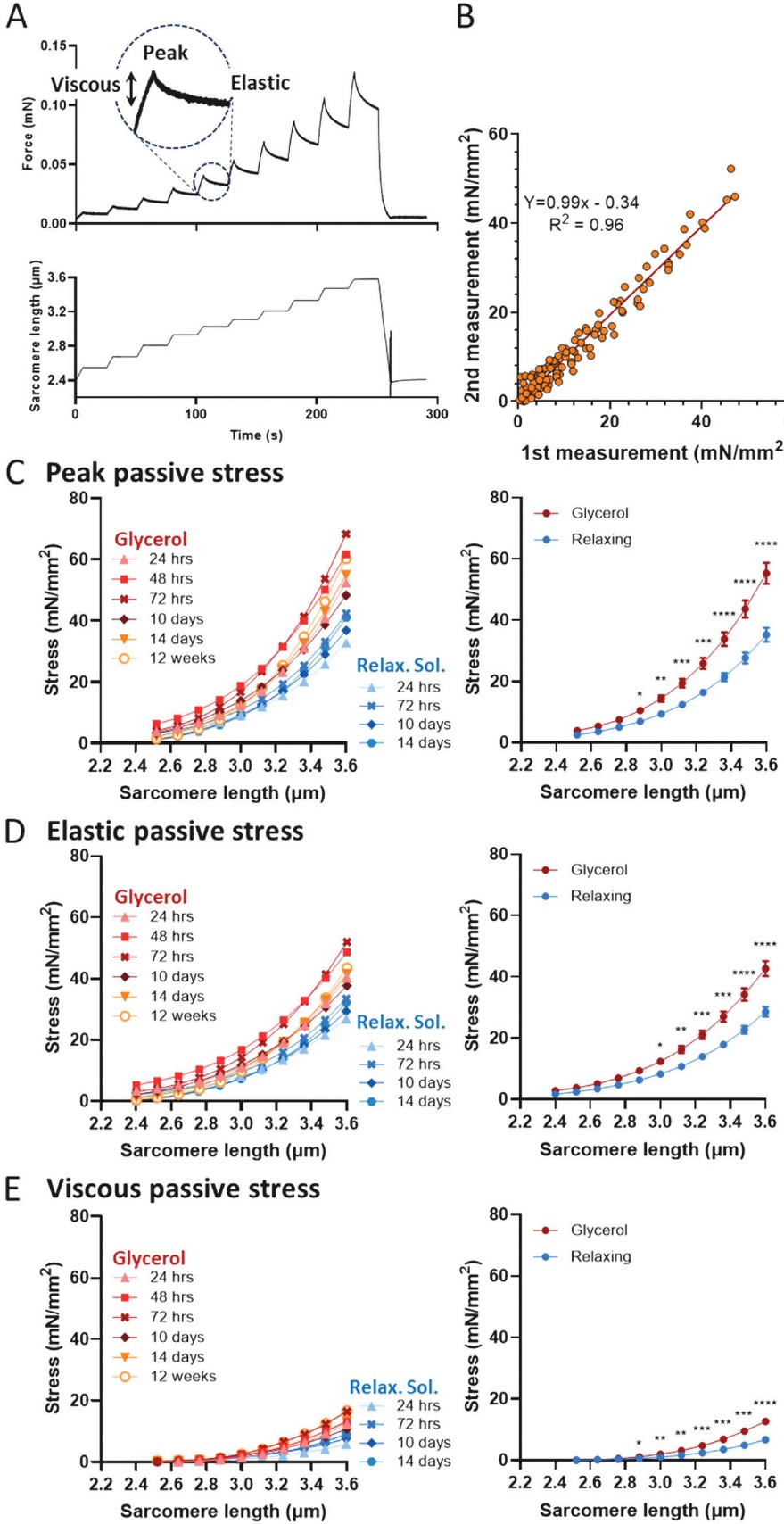

Figure 1. **Effect of glycerol storage on passive stress-SL relationship. (A)** Schematic of the stretch–hold protocol used, illustrating the definitions of peak, elastic, and viscous forces. Please note that elastic stress is in a quasi–steady-state stress at the end of the hold phase. **(B)** The reproducibility of the results by comparing two stretch–release cycles separated by a 10-min rest period at slack length (results from 10 fibers that have been stored for 3 days in glycerol). **(C–E)** Passive stress-SL relationships for fibers stored either in relaxing solution or glycerol for durations ranging from 1 to 14 days. The left panels show individual graphs (8–10 fibers per data point), while the right panels present the mean values for fibers stored in relaxing solution or glycerol. Glycerol storage leads to a statistically significant increase in peak stress, elastic stress, and viscous stress at SLs greater than ~2.8 μm. For detailed statistics, see Supplemental material.

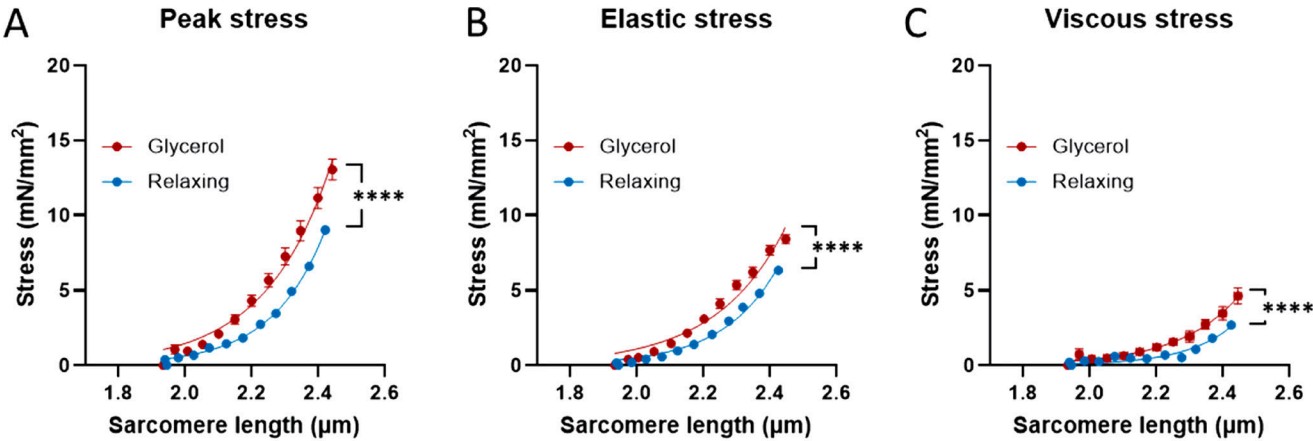

**Figure 2. Passive stress of cardiac myocytes is increased following glycerol storage. (A–C)** Changes in peak (A), elastic (B), and viscous (C) stress as a function of SL, when cardiomyocytes were skinned and stored in either relaxing solution at 4°C or glycerol at −20°C for 3–7 days. Significant increase in passive stiffness was observed in glycerol-stored (red curve; $n = 33$) cardiomyocytes compared with those of stored in relaxing solution (blue curve; $n = 15$). Extra-sum-of-squares F test; P < 0.0001.

(expressed in pCa, −log[Ca²⁺]), ranging from 4.5, 5.5, 5.75, 5.9, 6.0, 6.25, 6.5 to 9.0 pCa, with one force–pCa curve per fiber. Skinned muscle fibers were stored in either relaxing solution or glycerol for 3–5 days. Muscle fibers were washed in relaxing solution, mounted in the mechanical apparatus, and then set at a SL of either 2.6 or 3.0 µm and placed in pCa 9.0 first and then immerged in progressively higher calcium concentration once the force reaches a steady state at each pCa solution. The measured active force was converted to stress by dividing by the CSA of the fibers, and then normalized to its maximum stress at pCa 4.5. After curve fitting the stress–pCa data with a Hill-equation, the pCa₅₀ and Hill coefficient was determined.

### Thin filament removal
To eliminate potential contribution of the thin filament to effects of glycerol storage on passive stress, thin filaments were extracted using the actin-severing protein gelsolin (Granzier and Wang, 1993a; Trombitás et al., 2003). Following dissection, the EDL muscles were permeabilized overnight in skinning solution. 40 single fibers were then isolated and attached to T-clips. Half of the fibers were stored in relaxing solution, while the other half were treated with gelsolin (details provided in the Solution section) for 60 h. After treatment, the fibers were thoroughly rinsed with relaxing solution to remove residual gelsolin. The fibers were then divided into two groups: 10 fibers were stored in relaxing solution, and the remaining 10 fibers were stored in a glycerol/relaxing solution mix at −20°C. The storage duration for all fibers was 3–5 days.

### Methylglyoxal treatment
The EDL muscles were harvested and permeabilized overnight in skinning solution. Single muscle fibers were subsequently isolated and stored in either relaxing solution or a 50% glycerol/relaxing solution mix for 3 days. Following storage, the fibers underwent initial mechanical characterization before being treated with methylglyoxal solution (see Solution section) for 20 min. After treatment, the fibers were thoroughly rinsed with relaxing solution, and mechanical measurements were repeated.

### Mechanical evaluation using frozen-thawed muscle fibers
EDL muscle was rapidly frozen in liquid nitrogen, stored at −80°C for 5 years, transferred to −20°C for 1 h, and then thawed in skinning solution overnight (4°C). The muscle was thoroughly rinsed with relaxing solution and divided into two halves: one half was stored in relaxing solution (4°C), and the other half was stored in a 50% glycerol/relaxing solution mixture and kept at −20°C for 3–5 days. Passive stress of single fibers was then measured as described above.

### Solutions
Relaxing solution (pH 7.0): 40 mM BES, 10 mM EGTA, 6.56 mM MgCl2, 5.88 mM Na-ATP, 46.35 mM K-propionate, 15 mM creatine phosphate, 1 mM DTT, and protease inhibitors (1 mM E64, 1 mM leupeptin, and 1.25 mM PMSF) (Ogut et al., 1999). Activating solution (pH 7.0): 40 mM BES, 10 mM CaCO3-EGTA, 6.29 mM MgCl2, 6.12 mM Na-ATP, 45.3 mM K-propionate, 15 mM creatine phosphate, 1 mM DTT, and protease inhibitors (1 mM E64, 1 mM leupeptin, and 1.25 mM PMSF). Skinning solution: 1% Triton X-100 mixed added to the relaxing solution. Glycerol storage solution: 50% glycerol mixed with 50% of relaxing solution (assuming glycerol as a non-mixing additive, this results in a normal ionic strength relaxing solution). Gelsolin (0.5 mg/ml) in relaxing solution after reformulation to accommodate the ionic strength of MOPS storage solution containing gelsolin, and methylglyoxal solution: 200 µM methylglyoxal mixed with the relaxing solution.

### Statistics
All values are shown as mean ± SE. A two-way ANOVA with post hoc analysis was performed to detect significant difference between the muscle fibers that were stored in the relaxing solution and in the 50% glycerol at each SL, and the mixed-model ANOVA with a post hoc analysis was performed to detect difference in

active stress and $K_{tr}$ between the fibers stored in the two storage solutions for different duration. The extra-sum-of-squares F test was performed to examine changes in passive stress, and one-way ANOVA with Tukey's multiple comparison was performed to observe changes in the active stress in the four groups (i.e., glycerinated fibers versus non-glycerinated fibers, with and without gelsolin treatment on the fibers). The two-way ANOVA was performed to identify statistical difference in passive stress before and after the treatment with methylglyoxal for each SL, followed by a post hoc analysis. The level of significance shown in different number of asterisks: a $P < 0.05$ is shown as *, a $P < 0.01$ is shown as **, a $P < 0.001$ is shown as ***, and a $P < 0.0001$ is shown as ****.

### Online supplemental material

Fig. S1 shows the effect of storage solution when stored at the same temperature (4°C). Fig. S2 shows the fast and slow rates of force decay as a function of storage solution conditions before and after methylglyoxal treatment are shown. Datas S1 and S2 contain the original data for each figure and the statistical analysis of each figure.

## Results

### Effects of glycerol storage on passive stress

Passive stress was assessed using a stretch-hold, stepwise protocol as illustrated in Fig. 1 A. At the end of each stretch step, passive stress reached a peak, followed by a decline during the hold phase for 20 s due to the viscoelastic properties of muscle fibers. We measured both the peak stress and the quasi–steady-state stress at the end of the hold phase, referring to the latter for simplicity's sake as elastic stress, while the decline in stress during the hold phase was characterized as viscous stress (Fig. 1 A). This approach allowed us to construct relationships between SL and three passive stress parameters: peak stress, viscous stress, and elastic stress. Notably, the protocol did not induce any damage to the fiber, as evidenced by the reproducibility of the stress-SL relationships. Identical results were obtained when the stretch protocol was repeated after returning to the starting length and allowing a 10-min rest (Fig. 1 B).

Fig. 1, C–E (left panels), show passive stress–SL curves for fibers stored in either relaxing solution (4°C) or glycerol (–20°C) for various durations (1–14 days). Fibers stored in glycerol exhibited significantly higher passive stress values, including peak stress (Fig. 1 C), elastic stress (Fig. 1 D), and viscous stress (Fig. 1 E). This increase was evident after just 1 day of glycerol storage without progressive changes over time. In contrast, fibers stored in relaxing solution maintained stable passive stress curves with no noticeable deterioration over the 14-day study period. To summarize the findings, we averaged all curves for fibers stored in glycerol and those stored in relaxing solution, see Fig. 1, C–E, right panels. The results demonstrate significantly higher peak, elastic, and viscous stresses in glycerol-stored fibers (red curve). For instance, at an SL of 3.0 μm, the stress increase in the glycerol storage was 56.5 ± 13.6%.

To rule out potential effects of differences in storage temperatures (glycerol storage at –20°C versus storage in relaxing solution at 4°C), experiments were conducted using muscle fibers that were stored in either relaxing solution or glycerol with both kept at 4°C. Again, glycerol-stored fibers were significantly stiffer than those stored in relaxing solution (Fig. S1, A–C).

To address whether the effect of glycerol storage on passive stress is unique to skeletal muscle, we also studied cardiac myocytes, using a similar stretch-hold protocol as shown in Fig. 1 A, except that the SL range was adapted. Increased peak, elastic, and viscous stresses were observed in the glycerol-stored cardiomyocyte compared with those stored in the relaxing solutions (Fig. 2), indicating that glycerol storage increases passive stress of both skeletal and cardiac muscle.

### Effects of glycerol storage on active stress

Fibers were also activated, and the effect of storage conditions on maximal active stress was examined, along with measurements of $K_{tr}$. $K_{tr}$ reflects the rate at which crossbridges in muscle fibers reattach and generate force following disruption by a rapid shortening–re-lengthening maneuver, as illustrated in Fig. 3 A. This rate is determined from an exponential fit of the force redevelopment curve and provides insights into cross-bridge cycling kinetics.

As shown in Fig. 3 B, fibers stored in relaxing solution maintained stable maximal stress values (top) and unchanged $K_{tr}$ values (bottom) over time. Notably, fibers stored in glycerol exhibited results that were indistinguishable from those of fibers stored in relaxing solution. These findings indicate that the effects of glycerol storage are specific to passive stress, with no measurable impact on active stress or crossbridge cycling kinetics. Similar results were obtained when both storage solutions were kept at 4°C for 3–5 days, showing no changes in either active stress or $K_{tr}$ (See Fig. S1, D and E).

### Effects of glycerol storage on calcium sensitivity

To determine whether glycerol storage had an impact on calcium sensitivity of active force development, force–pCa relationships were measured at two SLs: 2.6 and 3.0 μm. At the short SL (2.6 μm), no effect on calcium sensitivity was found. However, at the longer SL (3.0 μm), there was a clear trend toward increased calcium sensitivity in glycerol-stored fibers (Fig. 4 B; $P = 0.085$). Moreover, when calculating the $\Delta pCa_{50}$ ($pCa_{50}$ at SL 3.0 μm minus $pCa_{50}$ at SL 2.6 μm), glycerol-stored fibers exhibited a significantly greater increase in calcium sensitivity (Fig. 4 C). This suggests that glycerol storage enhances the SL dependence of calcium sensitivity.

### Effect of glycerol storage on CSA of muscle fibers

If the CSA of muscle fibers were to be reduced by glycerol storage, passive stress would appear to be increased as a result. To identify possible morphological changes induced by glycerol storage, CSA was measured from all muscle fibers that were used for force measurements (see Materials and methods for details). The CSA of the glycerinated fibers was not different from the fibers stored in relaxing solution (Fig. 5), indicating that the observed increase in stiffness of the glycerinated fibers is not associated with changes in CSA of the fibers.

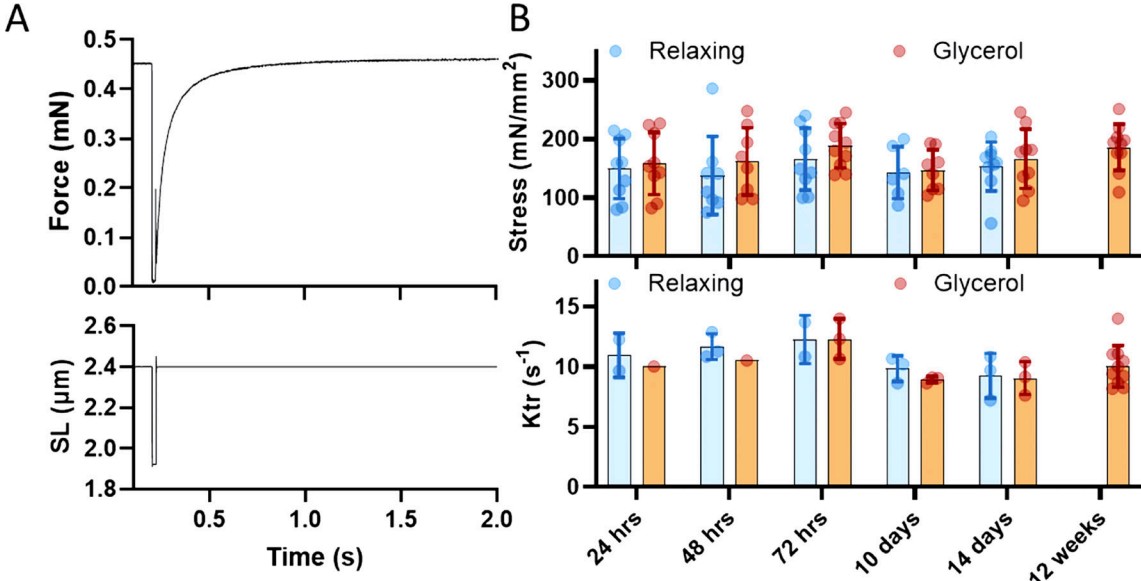

**Figure 3.** **Effect of glycerol storage on active stress and K$_{tr}$. (A)** Schematic representation of the K$_{tr}$ protocol. The fiber is maximally activated, and once a steady-state active force is achieved (top), the fiber undergoes a rapid release, followed by a rapid re-stretch (bottom trace shows the SL trace). During the release, force drops to zero and subsequently recovers after the re-stretch. The force recovery is fit with an exponential curve to determine the time constant, K$_{tr}$. **(B)** Top: Maximal active stress in fibers stored in relaxing solution (blue) or glycerol (brown) for various durations. Bottom: K$_{tr}$/s values. Glycerol storage has no effect on active stress nor K$_{tr}$.

### Effects of glycerol storage on thin filament–extracted muscle fibers

Whether the effect of glycerol storage on passive stress relies on the presence of the thin filament was also studied. Passive stress of single skeletal muscle fibers is largely determined by titin, but titin is known to weakly interact with the thin filament (Yamasaki et al., 2001). Thus, an enhanced titin–thin filament interaction could in principle explain the effect of glycerol

storage on passive stress. To examine this, fibers were treated with the actin-severing protein gelsolin, to remove the thin filaments. That gelsolin treatment is effective in removing thin filaments was revealed by the complete abolishment of active stress upon maximal activation in both fibers stored in relaxing solution and those stored in glycerol (Fig. 6 A). Thin filament–extracted fibers that were stored in glycerol for 3–5 days had significantly increased peak passive stress, elastic stress, and

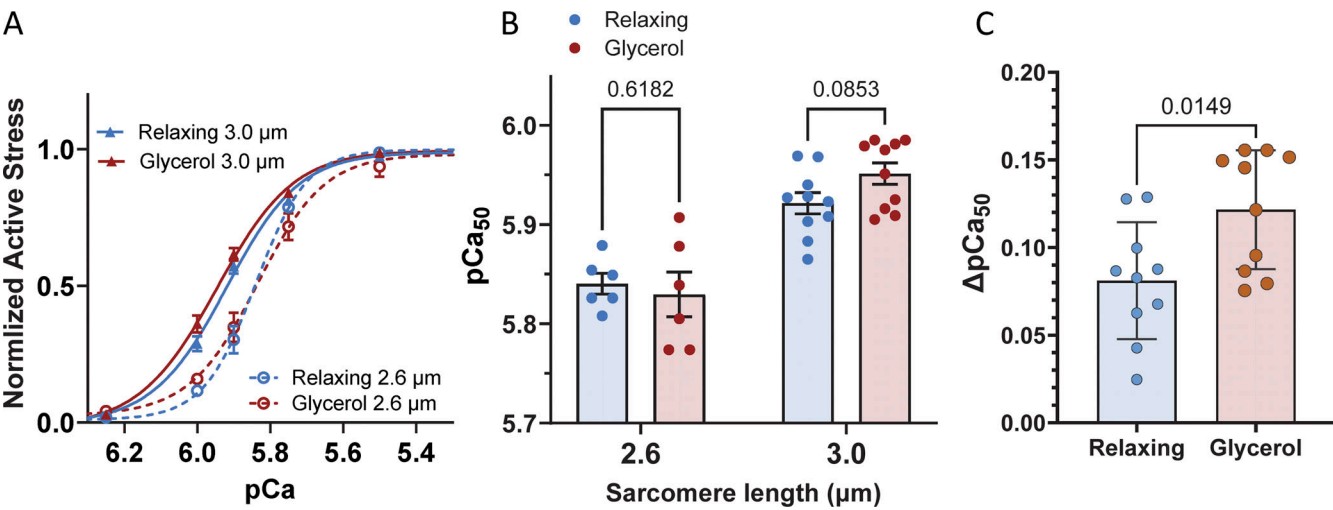

**Figure 4.** **Calcium sensitivity analysis of fibers stored in relaxing and glycerol solutions. (A)** Normalized active stress–pCa curves fitted to the data for each condition. The curves for both relaxing and glycerol storage solutions at a SL of 3.0 µm are left-shifted compared with those at SL 2.6 µm, indicating increased calcium sensitivity at the longer SL. **(B)** pCa$_{50}$ values for fibers stored in relaxing solution (blue) and glycerol solution (red) at SL 2.6 and 3.0 µm. No significant differences were observed between storage conditions at either SL, though there was a strong trend toward higher calcium sensitivity in glycerol-stored fibers at SL 3.0 µm. Two-way ANOVA; for details, see Supplemental material. **(C)** ΔpCa$_{50}$ (difference in pCa$_{50}$ between SL 3.0 µm and pCa$_{50}$ at SL 2.6 µm). Fibers stored in glycerol showed a greater increase in pCa$_{50}$ than those stored in relaxing solution (unpaired $t$ test).

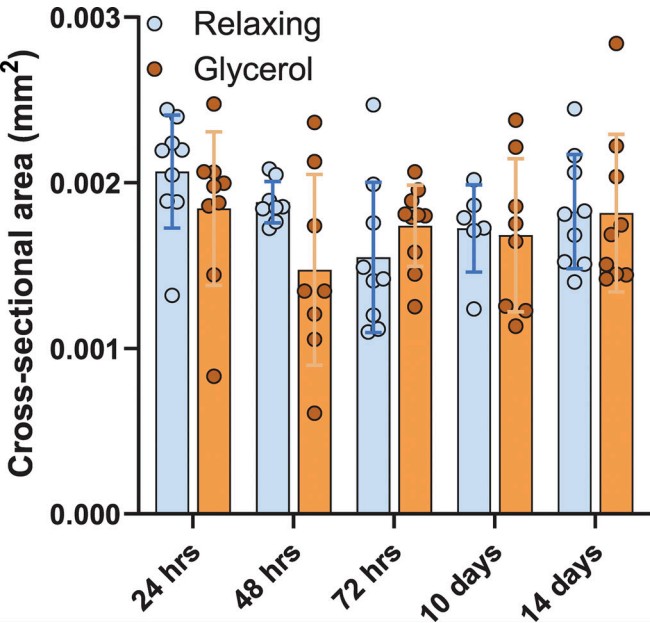

Figure 5. **Effect of glycerol storage on the CSA of fibers.** No change was observed in CSA of the fibers between the two different solution storages over time. Two-way ANOVA with Sidak's multiple comparisons.

viscous stress compared with those stored in the relaxing solution (Fig. 6, B–D). Thus, glycerol storage affects passive stress in a thin filament–independent manner.

### Effects of glycerol storage on muscle fibers treated with methylglyoxal

The increase in passive stress observed in glycerol-stored fibers occurs at SLs exceeding ~2.8 µm, a range where the extension of the PEVK region predominantly contributes to the overall extension of titin's I-band segment (Kellermayer et al., 1997). To investigate whether glycerol exerts its effect on passive stress by mechanisms that involve the PEVK segment, we conducted experiments using methylglyoxal, a reactive carbonyl compound and glycating agent that cross-links lysine residues via the Maillard reaction (Nagaraj et al., 1996). No change was observed in slack SL in muscle fibers before and after incubation with methylglyoxal, which were 2.367 ± 0.014 and 2.366 ± 0.016 µm, respectively (P = 0.65; paired $t$ test). However, treatment with methylglyoxal significantly increased the peak passive stress of fibers stored in relaxing solution (Fig. 7 A, left) as well as their elastic and viscous stress (Fig. 7 A, right). Interestingly, methylglyoxal treatment had no effect on fibers previously stored in glycerol, showing no change in peak passive stress (Fig. 7 B, left) or elastic and viscous stress (Fig. 7 B, right). These findings suggest that the modifications induced by methylglyoxal treatment and those resulting from glycerol storage involve the same target residues.

We also quantified the fast and slow rates of force decay and determined whether there are differences between the two storage solutions before and after methylglyoxal treatment. Interestingly, methylglyoxal treatment affected the fast rate of force decay of the fibers that were stored in the relaxing

solution, while no effect was observed in the glycerol-stored fibers. No change was observed in the slow rate of force decay for different storage solution with methylglyoxal incubation (see Fig. S2).

### Effects of glycerol storage on frozen muscle fibers

Recent research indicates that rapid freezing in liquid nitrogen, followed by storage at –80°C and then thawing in relaxing solution preserves ultrastructural and active properties of cardiac muscle (Milburn et al., 2022; Ma et al., 2023). To determine whether this storage method preserves passive stiffness of skeletal muscle, we studied long-term stored (5 years at –80°C) EDL muscle. Upon thawing in skinning solution and then washing with relaxing solution, one portion of the muscle was maintained in relaxing solution, while the other was stored in glycerol for 3–5 days (further details in Materials and methods). As shown in Fig. 8, passive stress in fibers frozen/thawed and then stored in relaxing solution (green triangles) was indistinguishable from that of fresh fibers (i.e., never frozen) and stored in relaxing solution (blue circles; adapted from Fig. 1, C–E). Furthermore, fibers that had been previously frozen and thawed exhibited increased passive stress after 3–5 days of glycerol storage (brown triangles)—behavior similar to that of fibers that had not undergone freezing but were stored in glycerol (red circles; adapted from Fig. 1, C–E). These findings suggest that rapid freezing followed by long-term storage at –80°C is a viable method for preserving muscle samples while maintaining their passive stress characteristics.

## Discussion

To investigate the molecular basis of muscle diseases, single-fiber mechanical studies are often employed. These fibers are typically obtained from muscles dissected from animal models or biopsies from patients. Given the labor-intensive nature of single-fiber studies, only a limited number of fibers can be analyzed per day. Consequently, long-term storage of muscles and biopsies is necessary. A commonly used storage method involves first demembranating the muscle tissues and then storing them in a glycerol-based solution, typically a 50% glycerol/relaxing solution mixture, at –20°C. It is generally assumed that this method preserves the mechanical properties of the tissues for at least several weeks. However, there has been limited investigation into this assumption, particularly regarding contractile dynamics and passive properties.

Our laboratory focuses on titin's passive stiffness and its alterations in disease. Anecdotal observations suggested that tissues stored in glycerol exhibited unexpectedly high passive stiffness, which motivated the present investigation. Notably, our findings revealed that glycerol storage for up to 2 wk did not impact key contractile properties, such as maximal active stress or the $K_{tr}$. Additionally, we found that storing demembranated fibers in a relaxing solution at 4°C effectively preserved their contractile characteristics throughout the 2-wk storage period, with replacing the relaxing solution every 24 h.

However, glycerol storage was observed to increase passive stress, affecting both its viscous and elastic components.

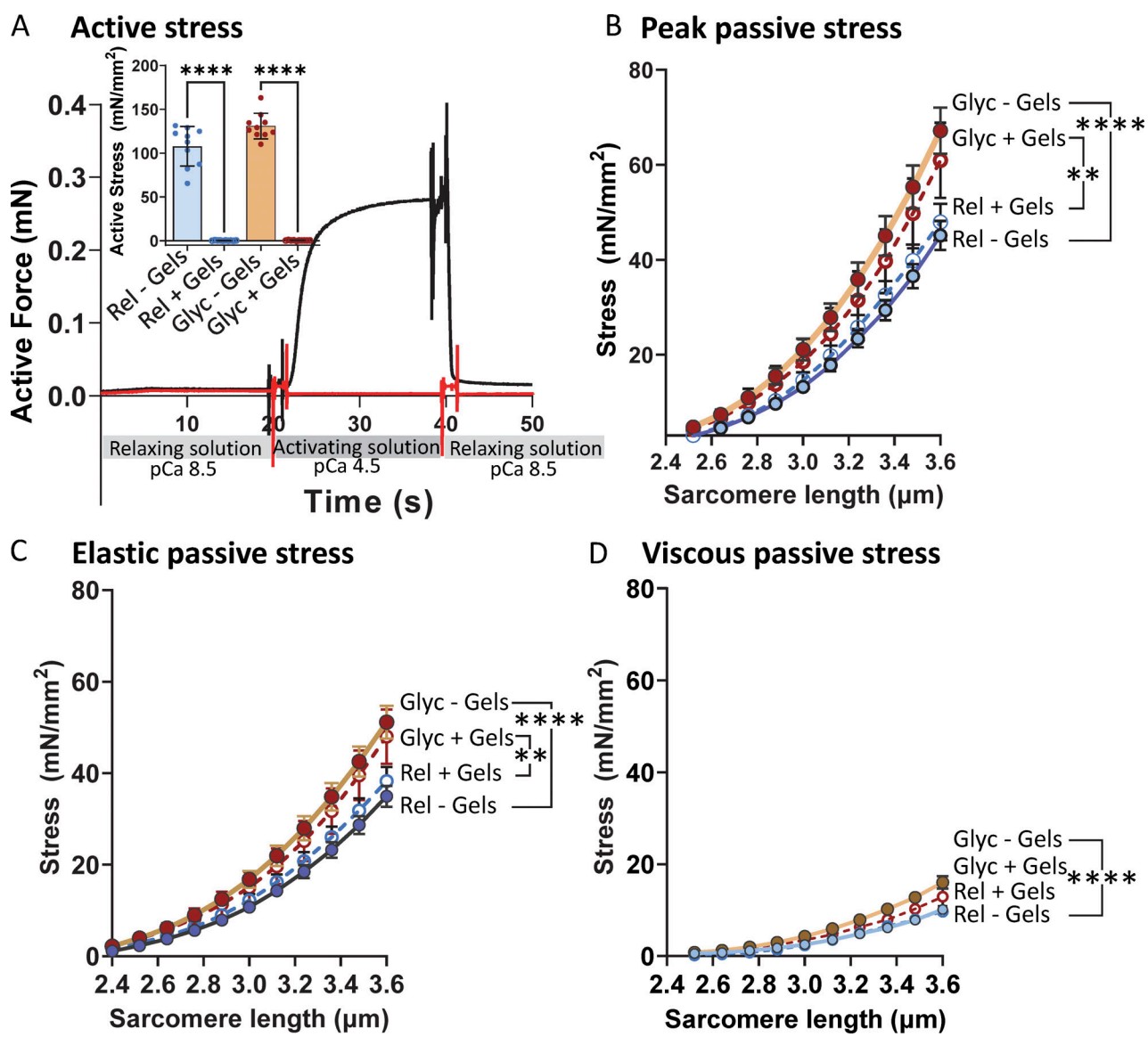

**Figure 6.** **Effect of glycerol on active and passive stress of thin filament–extracted muscle fibers. (A)** Gelsolin treatment fully abolishes active stress. Example of activation before (black trace) and after gelsolin treatment (red). Inset: Mean active stress in fibers that had been stored in relaxing solution and were then activated and some that were first gelsolin treated and then activated (blue symbols). The same experiment was performed on fibers that had first been stored in glycerol for 3 days (red symbols). Gelsolin treatment fully abolished active stress in fibers stored in relaxing solution or glycerol. **(B–D)** Passive stress of fibers that were gelsolin treated (open symbols) or not gelsolin treated (closed symbols) and then either stored in relaxing solution (blue symbols) or in glycerol (brown symbols) for 3 days followed by measurement of the passive stress–SL relations. Glycerol storage increases the peak, elastic, and viscous stress levels, irrespective of whether they were gelsolin treated. Extra-sum-of-square F test was run; 10 fibers per data point.

Mechanistic investigations indicate that these changes are likely due to molecular alterations within titin, particularly in its PEVK region. In the sections that follow, we provide a detailed discussion of these findings and their implications. Additionally, we offer recommendations for optimal tissue storage and highlight key insights gained from our study on methylglyoxal.

Titin is a giant filamentous protein that spans the half-sarcomere, extending from the Z-disk to the M-band (Fürst et al., 1989; Trombitás et al., 1998b). Its extensible I-band segment acts as an entropic spring, generating passive force when sarcomeres are stretched beyond their slack length. This force is the primary contributor to the passive stress within sarcomeres

(Granzier and Wang, 1993b) and constitutes a significant portion of the overall passive stress in skeletal muscle (Brynnel et al., 2018), where the extracellular matrix also plays a role (Meyer and Lieber, 2018; Ward et al., 2020).

Titin's force-extension behavior plays a vital role in several key muscle functions. It defines the SL range over which muscles operate (Brynnel et al., 2018), helps maintain the structural integrity of the contracting sarcomere (Horowits and Podolsky, 1987), ensures SL homogeneity along myofibrils (Granzier and Pollack, 1990; Granzier et al., 1991), and regulates contraction by straining the thick filament and modulating its OFF–ON transition (Irving et al., 2011; Ma et al., 2018, 2021). Additionally, it

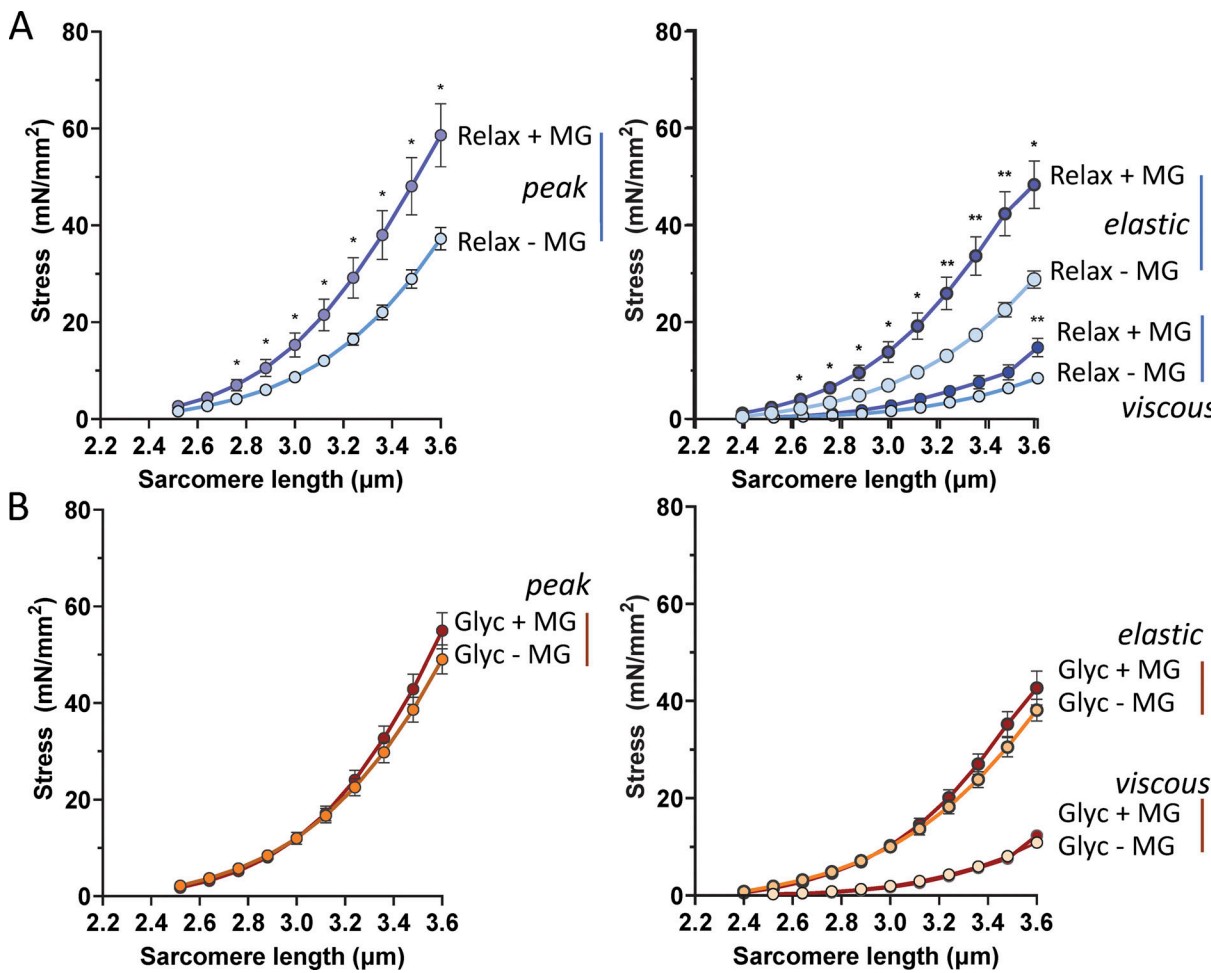

Figure 7. **Glycerol storage abolishes the effect of methylglyoxal on passive stress. (A and B)** Effect of methylglyoxal (MG) on passive stress of fibers stored in relaxing solution (A) or in glycerol (B). (Storage duration 4 days.) Left panel: Peak passive stress; right panels, elastic stress and viscous stress superimposed. MG increases passive stress of fibers stored in relaxing solution but has no effect on passive stress of fibers stored in glycerol. All values are shown as mean values of 10 fibers ± SE. Two-way ANOVA revealed significant changes only in the fibers that were stored in relaxing solution.

contributes to mechano-sensing pathways that drive hypertrophy signaling—particularly longitudinal hypertrophy (van der Pijl et al., 2018).

Studies have shown that titin's stiffness is often altered in both cardiac and skeletal muscle diseases (Hudson et al., 2011; Ottenheijm et al., 2012; van Hees et al., 2012; Lassche et al., 2013; Jeong et al., 2018; van der Pijl et al., 2019; Lin et al., 2022; Gohlke et al., 2024), sparking significant interest in understanding the mechanisms that regulate titin's stiffness and how these processes are disrupted in pathological conditions. In skeletal muscle, the extensible I-band segment of titin comprises several structural elements, including tandemly arranged Ig (Ig-like) domains, the N2A element (three Ig-like domains and several unique sequence insertions), and the PEVK region, rich in proline (P), valine (V), glutamate (E), and lysine (K) residues (Bang et al., 2001). The PEVK region is particularly critical for providing titin's elasticity within the physiological SL range of skeletal muscle (Trombitas et al., 1998b).

Titin's stiffness can be modulated through alternative splicing of its extensible I-band segment, which adjusts the number of Ig-like domains and the size of the PEVK region, altering the

protein's elastic properties (Cazorla et al., 2000, 2001; Freiburg et al., 2000; Bang et al., 2001). Additionally, stiffness can be fine-tuned posttranslationally. Phosphorylation of conserved sites within the PEVK region, mediated by protein kinases, such as PKCα and CaMKIIδ (Hidalgo et al., 2009, 2013; Anderson et al., 2010; Hudson et al., 2010; Hidalgo and Granzier, 2013), as well as acetylation of lysine residues within the PEVK region by lysine acetyltransferases (Lin et al., 2022), are key mechanisms for regulating titin's mechanical properties. These regulatory pathways highlight the dynamic nature of titin's stiffness and its potential dysregulation in disease require studies at the single muscle fiber level.

This study demonstrated significant increases in the passive stress of fibers stored in glycerol, with the effect becoming apparent after just 1 day and persisting throughout the studied 2-wk storage period (Fig. 1, C–E). To explore the underlying mechanism, one might hypothesize that glycerol storage has a compressive effect on the CSA of the muscle fiber. A reduction in CSA, without other changes, would be expected to increase passive stress, as stress is defined as the force divided by the fiber's CSA. However, a CSA reduction would also be expected to

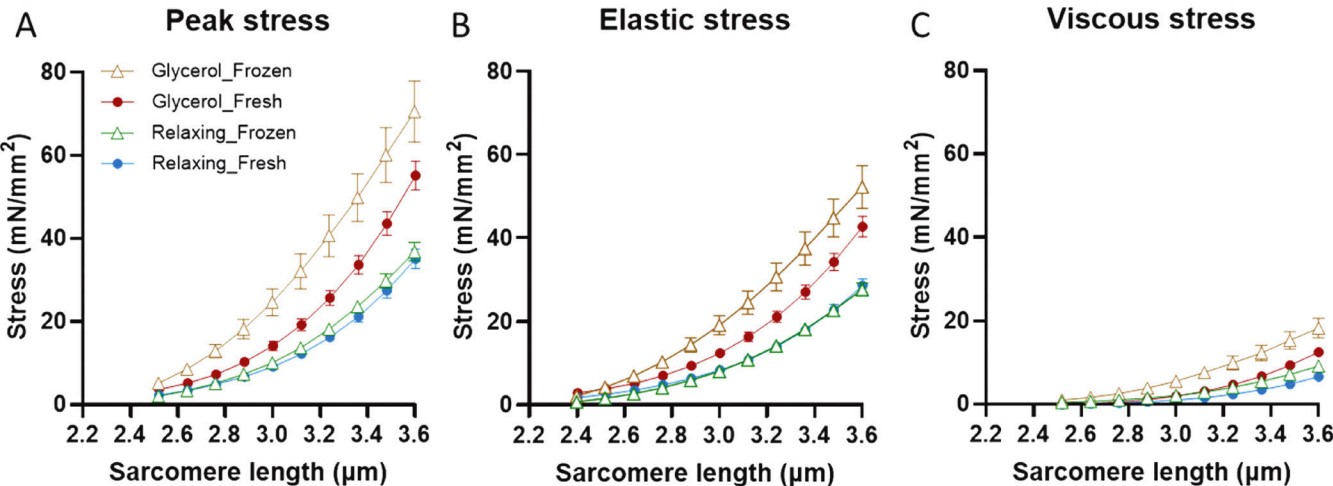

Figure 8. **Passive stress of fresh and frozen muscle that were stored in either relaxing solution or glycerol for ~5 days. (A–C)** Blue and red curves are the results of freshly skinned muscle fibers that were stored in relaxing solution and glycerol, respectively; adapted from Fig. 1, C–E. Green and brown curves are the results of frozen (5 years)/thawed muscle fibers, stored subsequently in either relaxing solution or glycerol, respectively. Note that the frozen fibers stored in relaxing solution (green curve) exhibited passive stress levels that were indistinguishable from those of fresh muscle stored in relaxing solution (blue curve).

increase active stress, which was not observed (Fig. 3 B). Furthermore, direct measurements of the CSA did not reveal any differences between the two storage methods (Fig. 5), suggesting that changes in passive stress were not due to alterations in fiber geometry.

An increase in passive stress could potentially arise from the formation of cross-links between the thin filament and titin. Since the PEVK element of titin is known to weakly interact with the actin filament (Kulke et al., 2001; Yamasaki et al., 2001; Linke et al., 2002; Chung et al., 2011), we considered the possibility that glycerol storage might enhance this interaction, leading to increased passive stress. To test this hypothesis, we used a calcium-independent recombinant gelsolin protein to remove thin filaments (Granzier and Wang, 1993b). Gelsolin is an actin-binding protein that severs actin filaments by binding near their barbed ends, disrupting interactions between adjacent actin subunits, and weakening filament integrity, ultimately fragmenting the filaments (Sun et al., 1999).

The effectiveness of gelsolin treatment was confirmed by the complete absence of active stress when the treated fibers were activated with maximal calcium (pCa 4.5), as shown in Fig. 6 A. However, despite the removal of thin filaments, glycerol-stored gelsolin-treated fibers exhibited elevated passive stress compared with fibers stored in a relaxing solution (Fig. 6, B–D). These results also rule out the possibility of enhanced thin–thick filament interactions, as the underlying mechanism for the increased passive stress observed with glycerol storage. Overall, the experiments with gelsolin support that the mechanism responsible for the elevated passive stress is intrinsic to titin.

Upon sarcomere stretch beyond the slack length, the tandem Ig segments initially dominate the extension of the I-band segment of titin, but at SLs greater than ~2.8 μm, however, the PEVK region primarily contributes to I-band segment extension (Trombitas et al., 1998b). Since the glycerol storage effect is prominent at SLs, where PEVK extension dominates (Fig. 1, B–D;

and Fig. 7 A), it is likely that the PEVK segment plays a key role in the underlying mechanism. Among the primary amino acids in the PEVK region—proline, glutamate, valine, and lysine—lysine stands out as the most reactive due to its nucleophilic primary amine group, which is involved in various chemical and biochemical reactions. To test lysine's involvement in the glycerol storage effect, we employed methylglyoxal, a reactive carbonyl and glycating agent known to form cross-links with lysine through the Maillard reaction (Nagaraj et al., 1996).

We observed that methylglyoxal significantly increased the passive stress of fibers stored in relaxing solution, with the effect being most pronounced at SLs greater than ~2.8 μm, where PEVK extension predominates. However, methylglyoxal treatment did not affect fibers previously stored in glycerol, as evidenced by the lack of change in peak passive stress (Fig. 7 B, left) or in elastic and viscous stress (Fig. 7 B, right). These findings suggest that the mechanisms underlying the modification induced by methylglyoxal treatment and glycerol storage might be similar. Specifically, the targeted region in titin for both methylglyoxal and glycerol is likely to be the PEVK region. However, future studies are required to address the detailed mechanism of how glycerol induces changes in the PEVK region.

The PEVK segment of skeletal muscle titin is large and rich in lysine, with 299 lysine residues in the mouse EDL PEVK, providing numerous opportunities for cross-linking (Table 1). The PEVK region is largely unfolded (Huber et al., 2012), allowing lysine pairs that are distant in the primary structure to regularly come into close proximity and form cross-links. Residues situated between cross-linked lysine pairs become mechanically "silent," meaning they do not contribute to stretch when the PEVK segment is extended. Consequently, the remaining "active" residues experience greater stretch, resulting in increased elastic stress. Thus, cross-linking of lysine residues provides a plausible explanation for the enhanced elastic stress observed in glycerol-stored fibers.

**Table 1. Lysine makes up a significant portion of the PEVK region (~15% of all amino acids) in three analyzed skeletal muscle types as well as in the left ventricle**

| Species | Region | Tissue | Lysine count | AA total | Percentage |
|---------|--------|--------|--------------|----------|------------|
| Mouse | N2B element | LV | 54 | 883 | 6.12% |
| Human | N2B element | LV | 64 | 928 | 6.90% |
| Mouse | PEVK | LV | 112 | 723 | 15.49% |
| Mouse | PEVK | EDL | 299 | 1,914 | 15.62% |
| Mouse | PEVK | Soleus | 284 | 1,816 | 15.64% |
| Mouse | PEVK | Diaphragm | 269 | 1,898 | 14.17% |

Lysine counts and percentages in the N2B region are much lower at 6–7% (see supplemental text at the end of the PDF). LV, left ventricle.

The increased viscous stress of fibers stored in glycerol may be attributed to multiple factors, including nonspecific cross-linking within the PEVK region, and the unfolding of Ig domains within the tandem Ig segments at high forces (Kellermayer et al., 2001; Trombitás et al., 2003). A straightforward explanation for the higher viscous stress is that glycerol storage increases elastic stress of muscle fibers and that this increases the likelihood of Ig domain unfolding, which results in greater viscous stress. Thus, we propose that the increased elastic stress amplifies viscous stress through this mechanism. Furthermore, because an increase in passive stress was observed in the glycerol-stored cardiomyocytes (Fig. 2) and the high percentage of lysine residues in the PEVK segment of mouse hearts (Table 1), it is likely that the proposed mechanism holds up for all isoforms.

The observation that methylglyoxal increases the passive stress of fibers stored in a relaxing solution (Fig. 7 A) is noteworthy. Methylglyoxal is a reactive dicarbonyl compound produced as a byproduct of glycolysis, primarily through the nonenzymatic breakdown of intermediates such as dihydroxyacetone phosphate and glyceraldehyde-3-phosphate (Gaens et al., 2013). It is well-known for contributing to cellular damage via the formation of advanced glycation end-products, which are associated with various pathological conditions (Gaens et al., 2013). Elevated levels of methylglyoxal have been observed in metabolic disorders, such as diabetes and obesity—key risk factors for cardiovascular diseases, including heart failure with preserved ejection fraction (Heinzel et al., 2020; Oliveira et al., 2024). Our novel finding that methylglyoxal increases passive stress in skeletal muscle fibers suggests an additional mechanism by which obesity and diabetes may contribute to musculoskeletal complications. Specifically, these conditions may promote increased skeletal muscle stiffness, adding to their overall pathological impact.

When skeletal muscle fibers were stored in glycerol for 4–8 h, no changes in passive stress were detected. However, a statistically significant increase was found in passive stress after as little as 24 h of glycerol storage (Fig. 1, C–E), ranging from 22 to 124% depending on SL, and the observed increase in stiffness was maintained over the full 12-wk testing period (Fig. 1, C–E). Similar findings were made in cardiac muscle (Fig. 2). Hence, we recommend that it is best to avoid glycerol for permeabilization and storage.

Would glycerol storage be acceptable if experimental and control samples (e.g., a biopsy from a patient with muscle disease and a biopsy from a healthy individual) are stored under the exact same conditions and durations? When the primary outcome is passive properties of muscle, the answer is no. If the experimental fibers have lysine modifications (e.g., acetylation or advanced glycation end-product cross-links), glycerol storage will not impact their passive stress, as seen with the lack of a glycerol storage effect in methylglyoxal-treated fibers (Fig. 7 B). In contrast, control samples without these lysine modifications will experience elevated passive stress due to glycerol storage. As a result, glycerol storage affects the two sample types differently, despite identical storage conditions. This discrepancy makes glycerol storage unsuitable for studying passive stress, as it introduces the potential for variability that compromises the comparisons.

Under our experimental conditions, glycerol storage effectively preserves the active contractile properties (maximal active stress and $K_{tr}$), indicating that the actomyosin system is insensitive to glycerol storage. This suggests that the impact of glycerol storage on muscle may be specific to titin, likely due to its unusually high lysine content and random coil structure. Additionally, calcium sensitivity of active stress was unaffected by glycerol storage at short SLs. However, the well-established increase in calcium sensitivity with SL (i.e., length-dependent activation) was more pronounced in glycerol-stored fibers compared with those stored in relaxing solution (Fig. 4 C).

A possible explanation lies in the role of titin-based passive stress in regulating the ON state of the thick filament (Granzier and Labeit, 2025). Previous studies on cardiac muscle have demonstrated that titin-based passive stress enhances calcium sensitivity (Fukuda and Granzier, 2005). The observation that glycerol storage increases passive stress in a SL-dependent manner—modest at SL 2.6 μm and substantial at SL 3.0 μm—supports this hypothesis. Therefore, while glycerol storage does not directly influence active stress, by increasing passive stress, glycerol indirectly affects the length dependence of activation.

In summary, our study demonstrates that glycerol storage effectively preserves the active contractile properties of skeletal muscle samples, including crossbridge cycling kinetics. However, glycerol storage alters the passive mechanical properties of muscle fibers, making it unsuitable for studies focusing on

these characteristics. Notably, our findings show that storing muscle fibers in a relaxing solution maintains stable passive and active stress responses for up to 2 wk, providing a practical time window for analyzing multiple fibers from a single sample. For scenarios where long-term storage is unavoidable, rapid freezing, storing at –80°C, and then thawing in relaxing solution might to be a promising alternative. Recent research suggests that this method preserves both the ultrastructural integrity and active properties of cardiac muscle (Milburn et al., 2022; Ma et al., 2023). Our studies on muscle, which had been stored long-term at –80°C for 5 years, demonstrated that passive stress remained indistinguishable from that of fresh fibers, while the glycerol storage effect persisted in prior frozen muscle. Therefore, glycerol storage should be avoided when assessing passive stiffness, but for long-term preservation, storage of rapidly frozen muscle at –80°C appears to be a viable option, if glycerol storage is avoided after the muscle has been thawed.

### Data availability
The data underlying all figures are available in the online supplemental material.

## Acknowledgments
Olaf S. Andersen served as editor.

We thank all lab members, in particular Drs. Johan Lindqvist and Robbert van de Pijl for providing support and scientific input.

This work was supported by the National Institutes of Arthritis and Musculoskeletal and Skin diseases R01AR083233. Open Access funding provided by the University of Arizona.

Author contributions: S.-W. Han: data curation, formal analysis, investigation, methodology, validation, visualization, and writing—original draft, review, and editing. J. Kolb: conceptualization, investigation, methodology, and writing—review and editing. G.P. Farman: conceptualization, data curation, formal analysis, and writing—review and editing. J. Gohlke: data curation, formal analysis, and writing—review and editing. H.L. Granzier: conceptualization, funding acquisition, project administration, supervision, validation, and writing—original draft, review, and editing.

Disclosures: The authors declare no competing interests exist.

Submitted: 23 November 2024

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

# Supplemental material

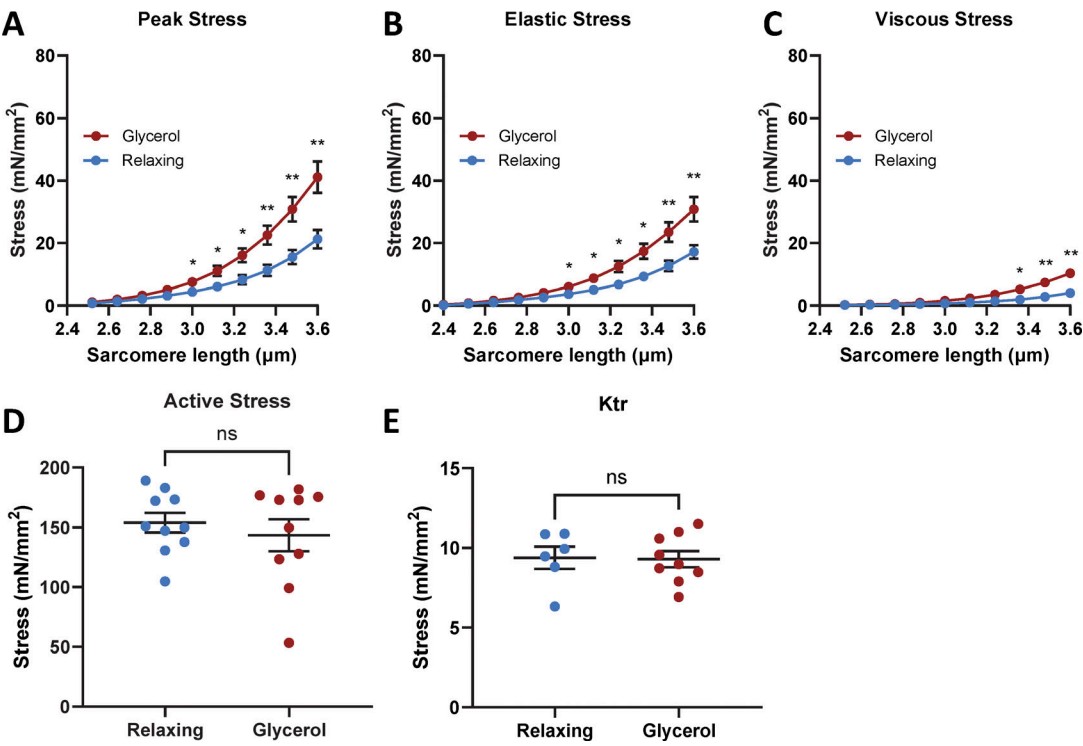

Figure S1.  **Glycerol storage at 4°C increases passive stress. (A and C)** Changes in peak (A), elastic (B), and viscous (C) stress as a function of SL, when muscle samples were stored at 4°C for both storage solutions. No change was observed in active stress (A) and rate of force redevelopment ($K_{tr}$; B) when both glycerinated and non-glycerinated fibers were stored at 4°C. **(A–E)** Two-way ANOVA with Sidak's multiple comparisons test, D and E, unpaired *t* test.

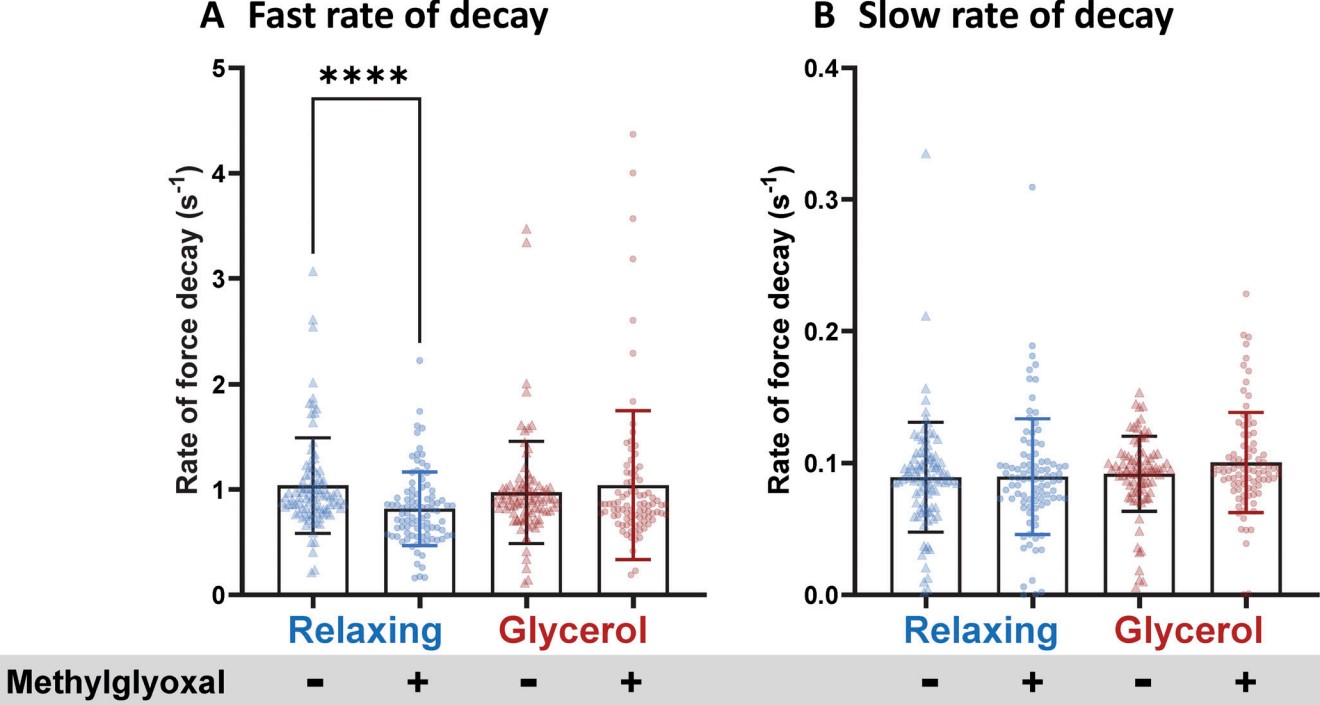

Figure S2.  **Fast and slow rates of force decay as a function of storage solution conditions before and after methylglyoxal treatment are shown. (A and B)** Significant reduction in the fast rate of force decay was observed in methylglyoxal treated fibers that had been stored in the relaxing solution (A, paired *t* test, P < 0.0001), while there was no change observed in the glycerol-stored fibers.

**Provided online are Data S1 and Data S2. Data S1 contains the original data for each figure. Data S2 provides the statistical analysis of each figure.**

*Methods*

*Calculation of lysine residues*

For calculation of lysine and total amino acid counts, the number of amino acids per exon was multiplied with exon inclusion levels from previous RNA-seq studies of LV, EDL, diaphragm, and soleus (Brynnel et al., 2018; Strom et al., 2024). Consequently, numbers and percentages should be representative of titin isoforms found in respective muscle types. For the N2B region, lysine and total amino acid counts are based on the complete N2B sequence because the region is fully included in left ventricular tissue.

