## [Peer Review File · The Journal of General Physiology]

Glycerol storage increases passive stiffness of muscle fibers through effects on titin extensibility.

Henk Granzier, Seong-won Han, Justin Kolb, Gerrie Farman, and Jochen Gohlke

Corresponding Author(s): Henk Granzier, University of Arizona Medical Center

Review Timeline:

Submission Date:	November 23, 2024
Editorial Decision:	December 19, 2024
Revision Received:	March 25, 2025
Editorial Decision:	April 8, 2025
Revision Received:	April 18, 2025

Editor: Olaf Andersen

Transaction Report:

DOI: <https://doi.org/10.1085/jgp.202413729>

December 20, 2024

Dr. Henk L. Granzier
University of Arizona Medical Center
Dept. of Cellular and Molecular Medicine
1656 East Mabel Street, MRB Room 325
Tucson, AZ 85724-5217

Re: 202413729

Dear Dr. Granzier,

Thank you for submitting your manuscript, entitled "Glycerol storage increases passive stiffness of skeletal muscle fibers through effects on the extensibility of titin's PEVK segment." to JGP. Your manuscript has now been seen by 3 reviewers, whose comments are appended below. You will see that the reviewers consider this to be an important study. They identify a number of issues for your consideration, some of them call for an additional experimental effort, others call for revisions of the text. Reviewer #1's comment 1d, which is echoed by the other reviewers, f. ex., goes to what I consider to be a key point; addressing/resolving it will require considerable skill and diplomacy. Given the prevalent use of glycerol storage in studies on muscle mechanics, your results will become an important source of information for future studies; it also will force reexamination of many previous studies from many laboratories.

We hope that you will be able to submit a revised manuscript that addresses the letter and the spirit of the reviewers' comments. We will request that the original reviewers evaluate the manuscript. In addition, please do not hesitate to contact me (via the editorial office) if you feel that a discussion of the reviewers' and editors' comments would be helpful.

Please submit your revised manuscript via the link below, along with a point-by-point letter that details your response to the reviewers' and editors' comments, as well as a copy of the text with alterations highlighted (boldfaced or underlined). If the article is eventually accepted, it would include a 'revised date' as well as submitted and accepted dates. If we do not receive the revised manuscript within one year, we will regard the article as having been withdrawn. We would be willing to receive a revision of the manuscript at a later time, but the manuscript will then be treated as a new submission, with a new manuscript number.

Please pay particular attention to recent changes to our instructions to authors in the following sections: Data presentation, Blinding and randomization and Statistical analysis, under Materials and Methods, as shown here: <https://rupress.org/jgp/pages/submission-guidelines#prepare>. Re-review will be contingent on inclusion of the required information (including for data added during revision) and demonstration of the experimental reproducibility of the results. Also, To improve the reproducibility of published content, we have partnered with SciScore. Authors are prompted in eJP to copy and paste the Materials and Methods section of their manuscript for a SciScore assessment when submitting their revised manuscript. Authors are encouraged (not required) to further revise their Materials and Methods if the SciScore is below 4. More information can be found here: <https://rupress.org/jgp/pages/submission-guidelines#sciscore>.

Please note, JGP now requires authors to submit Source Data used to generate figures containing gels and Western blots with all revised manuscripts (when applicable). This Source Data consists of fully uncropped and unprocessed images for each gel/blot displayed in the main and supplemental figures. If your paper includes cropped gel and/or blot images, please be sure to provide one Source Data file for each figure that contains gels and/or blots along with your revised manuscript files. File names for Source Data figures should be alphanumeric without any spaces or special characters (i.e., SourceDataF#, where F# refers to the associated main figure number or SourceDataFS# for those associated with Supplementary figures). The lanes of the gels/blots should be labeled as they are in the associated figure, the place where cropping was applied should be marked (with a box), and molecular weight/size standards should be labeled wherever possible. Source Data files will be made available to reviewers during evaluation of revised manuscripts and, if your paper is eventually published in JGP, the files will be directly linked to specific figures in the published article.

Source Data Figures should be provided as individual PDF files (one file per figure). Authors should endeavor to retain a minimum resolution of 300 dpi or pixels per inch. Please review our instructions for export from Photoshop, Illustrator, and PowerPoint here: <https://rupress.org/jgp/pages/submission-guidelines#revised>

Whilst you are revising your manuscript, we ask that you consider whether you have any artwork that might be suitable for the cover of JGP. Microscopy images are particularly good for cover artwork, but other types of image can be very effective, so we encourage you to be creative. Please don't restrict yourself to images from the paper; an image that is relevant to the work described would be just as suitable. Images should be a minimum resolution of 300 dpi. To see recent examples, visit the following page and click on 'Show covers? Yes': <https://jgp.rupress.org/content/by/year>

Thank you for submitting your interesting research to JGP.

Please submit your revised manuscript, and any associated files, via this link:
Link Not Available

Sincerely,

Olaf S. Andersen, M.D.
On behalf of Journal of General Physiology

Journal of General Physiology's mission is to publish mechanistic and quantitative molecular and cellular physiology of the highest quality; to provide a best-in-class author experience; and to nurture future generations of independent researchers.

Reviewer #1 (Comments to the Authors):

The manuscript, "Glycerol storage increases passive stiffness of skeletal muscle fibers through effects on the extensibility of titin's PEVK segment" by Han et al evaluates differences in passive tension and stiffness of skeletal muscles stored for up to 14 days in the presence or absence of glycerol. The authors show that permeabilized (skinned) mouse skeletal (ED) muscles increase in passive tension in the presence of glycerol. Further, this appears to be a glycation induced change, as the addition of methylglycol will induce stiffening in a muscle stored without glycerol, but no change in muscles stored in glycerol.

The implication of this finding in comparisons of muscles is striking, especially given that the storage of muscle fibers is a widely utilized technique that can improve the reduction and reuse of tissues collected from pre-clinical models and potential human samples. Given the importance of titin-based passive tension previously reported by the author(s) and other investigators, this confounding change in tension and stiffness is important.

While the data presented is generally convincing, this reviewer provides several comments that may enhance confidence in the results and better understanding of how these results may impact our acceptance of previously published studies.

Major Comments

1. As the author's laboratory has published multiple articles utilizing glycerol storage and several other laboratories use glycerol in muscle fibers, a more complete understanding of the study's implication may be warranted.
 - 1a. Since the increased passive stiffness effect is thought to be generated from the PEVK region and the PEVK region is highly differentially spliced, do the authors have a prediction (such as WLC model and/or evidence) of fibers sourced from other muscle groups that would be impacted?
 - 1b. The EDL muscles have a reasonably consistent number of PEVK and Ig domains. Can the authors fully discount the potential for Ig domain unfolding to contribute to the viscous effect at high sarcomere lengths (perhaps via different stretch speeds)? If the authors cannot fully discount Ig domain unfolding, consideration in the discussion may be added as this effect may similarly be important when considering fibers from other muscle types.
 - 1c. Glycerol is sometimes used for permeabilization of muscles, including cardiac muscles. A discussion of the implications of this use would benefit the study. i.e. How much impact would the glycerol permeabilization have if just used for 24 hrs? How much of an impact would glycerol exposure have on cardiac muscles, especially since dysfunctional hearts typically express greater content of N2BA (e.g., diabetes induces an increase in N2BA element content) and thus have greater PEVK content?
 - 1d. Importantly, the discussion lacks guidance on whether prior work can generally be trusted or if studies mentioning glycerol in their methods need to be replicated using glycerol-free solutions specifically. What model systems might be more reliable than others?
2. Differences in Storage Conditions
There samples stored in glycerol are stored at -20 degrees C, while samples stored without glycerol are stored at 4 degrees C. The study would be improved with a control for storage temperature (i.e. glycerol at 4 degrees C).
3. Changes in Cell Morphology
 - 3a. Slack sarcomere lengths should be reported to better understand whether the glycation is solely influencing the stretch or whether the slack properties may be changed.
 - 3b. Supplemental Figure 1 (Fiber CSA), should also be described in the results.
4. Effect on Active Stress
The authors describe no change in active stress or ktr. However, investigators have noted changes in EC₅₀ with the addition of methylglyoxal (PMID 30333300). A consideration of the limitation that other changes in calcium sensitivity (independent of active

stress and ktr) would improve the discussion.

The authors may wish to include some context relative to whether the change in passive stiffness may contribute to the explanation, but should be cautious not to attribute all change to titin without evidence.

5. Statistics

5a. It is unclear why the investigators used multiple T-tests instead of ANOVA for conditions that included sarcomere length and storage conditions. Please clarify this reasoning

5b. This reviewer suggests the inclusion of all F- and p-values, perhaps in the supplement. Inclusion would improve this reviewer's understanding of whether main effects or interactions are significant.

5c. Sample size is noted on some figures as 8-10 fibers. The text should clearly indicate whether these are biological or technical replicates.

Minor Comments:

6. Characterization of Viscous Stress

6a. Was the viscous stress calculated strictly from the 20-second hold? Because the viscous element is clearly not completely removed, does this severely underestimate the viscous stresses? An estimation of the actual steady-state elastic stress via the plateau of a single or double exponential may be of value.

6b. Discussion, e.g. page 13. The authors indicate that PEVK glycation is likely present and show that stretch-relaxation of a muscle alters the fibers after step-stretches. Would the authors predict a different result if using the protocol described in Nedrud Biophys J 2011 (PMID 21943419)?

7. Methods- Solutions

The methods describe 1% Triton X-100 and 50% glycerol. For each, do either addition dilute the relaxing solutions? (For example, is 1.01 mL 100% triton added to 100 mL of relaxing solution or is a concentrated relaxing solution stock used? For glycerol storage, is the glycerol considered a component of the solution or a non-mixing additive making a 1:1 mix of glycerol and normal concentration relaxing solution?)

Reviewer #2 (Comments to the Authors):

Dear authors,

The study addresses a longstanding issue, whether glycerol storage has unintended effects on the mechanical properties of skeletal (and possibly cardiac) muscle fibers, and the answer is yes. The authors also provide a mechanism and test this. They point out that these phenomenon may occur in metabolic syndromes as well. It is therefore not only methodologically important but may also have clinical implications. The paper is neat and straightforward. It is definitively an excellent contribution to the field. There are a few minor comments.

At the top of page 9, there is a paragraph that seems to contradict the rest of the paper, in particular this sentence: "Thin-filament extracted fibers had significantly increase peak passive stress, elasticstress and viscous stress (Figs. 3B-D)." My guess is that this is just an editing mistake, but the authors should correct this.

The second comment is on the statistics of how the authors compare two curves. They divide the curve up in discrete steps, and compare the steps using multiple t-tests. The graphs in figure 1c-e are an example. There is probably some binning of the values on the x-axis as well, which will affect the variation. While there clearly is a difference between the curves, no argument there, I am fairly sure that this is not the appropriate way to do it. The research group has used this method for many years, and it is visually clear, but not correct. If you do this many t-test and properly correct for multiple comparisons (which I don't think the authors have done for the passive force curves), the bar for significance is really high, and you lose a lot of statistical power. I don't have the exact answer of how to do it, but for comparing two curves you normally use regression techniques, so all the xy pairs of the curves are properly weighted, the correlation between the points on the curve is taken into account and therefore the statistical power increases greatly. It would also give you confidence intervals instead of error bars. It is however a minor issue for this paper and not essential to its conclusions.

Reviewer #3 (Comments to the Authors):

The manuscript by Han et al investigates the effect of glycerol storage on the passive mechanical properties of skeletal muscle fibers, emphasizing the role of titin's PEVK segment. It finds that while glycerol storage preserves active force and crossbridge cycling kinetics, it significantly increases passive tension in fibers due to titin-specific alterations. The research connects these

findings to potential pathological implications in metabolic disorders, as similar effects are noted with methylglyoxal-induced lysine crosslinking.

Storing muscle and muscle-protein samples in glycerol has been a commonly used method since the ground-breaking era of muscle research in the mid-twentieth century. It has been essentially taken for granted, albeit never tested critically, that striated-muscle structure and function are preserved by this method, which has since been widely employed. The current work thus executes a highly timely test of this age-old hypothesis. The methods are robust, employing standard techniques such as mechanical testing of isolated muscle fibers, gelsolin treatment for thin filament removal, and methylglyoxal exposure to mimic glycation-type modifications. Statistical rigor is high, the large sample size is to be commended, and the tests are executed and interpreted according to the state of the art. The results are discussed, albeit with caution, in the context of potential clinical significance. Altogether, this is an important and straight-forward work.

Major comments:

1. The authors analyze the passive muscle-mechanics data in terms of three parameters: peak force, "quasi-steady-state" stress ("elastic stress", measured after 20 seconds of force relaxation), and the difference between them ("viscous stress"). While such a partitioning of data allows for comparison, the terminology should be cautiously used, and the analysis should be carefully discussed. First, such a "passive" mechanical system approaches equilibrium (not steady state), unlike active isometric tension. Therefore, the "quasi-equilibrium stress" may be more appropriate. Second, it should be kept in mind that the viscous response of the muscle fiber occurs throughout the stretch-and-hold measurement process, thus it alters the peak passive force and it would proceed beyond the 20-second (i.e., the equilibrium is far from set). The arbitrarily chosen 20-second examination window allows for precise comparisons only if the rate of the force decay within this time window is constant. This could actually be easily tested and even used for measuring the glycerol-storage effect. Thus, I recommend to show a comparison of the force decay (for relaxing, glycerol and methylglyoxal situations), and to discuss the above considerations.

2. The authors claim that the glycerol-storage effect is more prominent towards longer sarcomere lengths. While the gap between the control and glycerol curves is indeed widening towards longer sarcomere lengths, notably the entire dataset curves upwards. It would be an easy test to divide the two curves by each other. This way the above claim could be experimentally tested.

3. The glycerol storage effect is slightly lowered after gelsolin treatment (Figure 3). What might be the reason for it?

4. In a very nice experiment (Figure 4) the authors show that methylglyoxal treatment of the muscle fiber essentially mimics the glycerol storage effect. I agree that it is tempting to claim therefore that glycerol acts in the same way as methylglyoxal, but I find it premature, simply because glycerol is not a chemical cross-linker. Thus, while the target of glycerol and methylglyoxal might be identical, their mechanisms of action may still be different. The authors may wish to discuss this possibility.

Minor comments:

1. Using abbreviation in the abstract should be avoided if possible (see EDL).

2. The "tr" (tension recovery) in K_{tr} should be in the subscript throughout the manuscript.

3. Page 7, second paragraph, first sentence: "... (left panels) show passive..." (not "shows").

4. On page 12, in the third paragraph the authors raise the hypothetical possibility that glycerol treatment may induce a compressive effect on the muscle fiber. By what mechanism? Osmosis?

5. Page 14, last paragraph, first sentence: I recommend complementing as "...storage effectively preserves the active contractile properties..."

Comments by Dr Olaf Andersen, JGP Editor: *“You will see that the reviewers consider this to be an important study. They identify a number of issues for your consideration, some of them call for an additional experimental effort, others call for revisions of the text. Reviewer #1's comment 1d, which is echoed by the other reviewers, f. ex., goes to what I consider to be a key point; addressing/resolving it will require considerable skill and diplomacy. Given the prevalent use of glycerol storage in studies on muscle mechanics, your results will become an important source of information for future studies; it also will force reexamination of many previous studies from many laboratories. “*

Response: Thank you for your support in reviewing our manuscript. Below, we have addressed each of the reviewers' comments point by point, including those mentioned above. All revisions made to the manuscript are highlighted in yellow.

Reviewer #1 (Comments to the Authors):

Comment: *“The manuscript, “Glycerol storage increases passive stiffness of skeletal muscle fibers through effects on the extensibility of titin's PEVK segment” by Han et al evaluates differences in passive tension and stiffness of skeletal muscles stored for up to 14 days in the presence or absence of glycerol. The authors show that permeabilized (skinned) mouse skeletal (ED) muscles increase in passive tension in the presence of glycerol. Further, this appears to be a glycation induced change, as the addition of methylglycol will induce stiffening in a muscle stored without glycerol, but no change in muscles stored in glycerol. The implication of this finding in comparisons of muscles is striking, especially given that the storage of muscle fibers is a widely utilized technique that can improve the reduction and reuse of tissues collected from pre-clinical models and potential human samples. Given the importance of titin-based passive tension previously reported by the author(s) and other investigators, this confounding change in tension and stiffness is important.” “While the data presented is generally convincing, this reviewer provides several comments that may enhance confidence in the results and better understanding of how these results may impact our acceptance of previously published studies.”*

Response: Thank you for your positive review of our work and for your insightful comments, which have helped us further strengthen our paper.

Major Comments:

Comment: *“1. As the author's laboratory has published multiple articles utilizing glycerol storage and several other laboratories use glycerol in muscle fibers, a more complete understanding of the study's implication may be warranted.*

1a. Since the increased passive stiffness effect is thought to be generated from the PEVK region and the PEVK region is highly differentially spliced, do the authors have a prediction (such as WLC model and/or evidence) of fibers sourced from other muscle groups that would be impacted?”

Response: Based on the methylglyoxal study we hypothesized that the mechanism underlying the glycerol effect involves lysine residues in the PEVK region. Using RNA-seq data previously collected from various mouse muscle types, we analyzed the composition of the PEVK segment across different titin isoforms. This analysis revealed a high percentage of lysine residues in the PEVK segment of all isoforms, including N2B element, and that this is unique to the PEVK segment (see Table 1 below). Therefore, if the glycerol effect indeed depends on lysine residues in the PEVK region, all isoforms are expected to become stiffer following glycerol storage.

Table 1

Species	Region	Tissue	Lysine count	AA total	Percentage
mouse	N2B element	LV	54	883	6.12%
human	N2B element	LV	64	928	6.90%
mouse	PEVK	LV	112	723	15.49%
mouse	PEVK	EDL	299	1914	15.62%
mouse	PEVK	Soleus	284	1816	15.64%
mouse	PEVK	Diaphragm	269	1898	14.17%

To experimentally test this conclusion, we examined the effect of glycerol storage on passive stress of mouse left ventricular cardiomyocytes. Cardiomyocytes stored in glycerol exhibited increased passive stress compared to those stored in relaxing solution (see Figure 1 below), supporting that the glycerol storage effect is not isoform-specific but rather universal. In the revised manuscript, we have added Methods (page 5-6, line 128-136), Results (page 9, line 240-250, Figure 2), and Discussion (page 17, line 538-540)

Figure 1. Changes in peak (A), elastic (B), and viscous (C) stress as a function of sarcomere length of cardiomyocytes stored in either relaxing solution or glycerol for 3-5 days. Significant increases in passive stress were observed in glycerol-stored cardiomyocytes (red curve; n=33) compared to those of stored in relaxing solution (blue curve; n=15). Extra Sum-of-Squares F-test; $p < 0.0001$

Comment: “1b. The EDL muscles have a reasonably consistent number of PEVK and Ig domains. Can the authors fully discount the potential for Ig domain unfolding to contribute to the viscous effect at high sarcomere lengths (perhaps via different stretch speeds)? If the authors cannot fully discount Ig domain unfolding, consideration in the discussion may be added as this effect may similarly be important when considering fibers from other muscle types.”

Response: We agree with the reviewer’s comment, and in the original manuscript had the following paragraph: “The increased viscous stress of fibers stored in glycerol may be attributed to multiple factors, including non-specific crosslinking within the PEVK region, and the unfolding of Ig domains within the tandem Ig segments at high forces (Kellermayer et al., 2001; Trombitas et al., 2003). A straightforward

explanation is that the enhanced elastic stress caused by PEVK crosslinking that occurs during glycerol storage increases the likelihood of Ig domain unfolding or the breakage of non-specific labile crosslinks, thereby contributing to greater viscous stress. Thus, we propose that the increased elastic stress amplifies viscous stress through this mechanism.”

In the revised manuscript, we edited this statement to make the paragraph clearer (page 17, line 530-537), and now it reads: *“The increased viscous stress of fibers stored in glycerol may be attributed to multiple factors, including non-specific crosslinking within the PEVK region, and the unfolding of Ig domains within the tandem Ig segments at high forces (Kellermayer et al., 2001; Trombitas et al., 2003). A straightforward explanation for the higher viscous stress of fibers stored in glycerol is that glycerol storage increases elastic stress of muscle fibers and that this increases the likelihood of Ig domain unfolding, which results in greater viscous stress. Thus, we propose that the increased elastic stress amplifies viscous stress through this mechanism.”*

Comment: *“1c. Glycerol is sometimes used for permeabilization of muscles, including cardiac muscles. A discussion of the implications of this use would benefit the study. i.e. How much impact would the glycerol permeabilization have if just used for 24 hrs? How much of an impact would glycerol exposure have on cardiac muscles, especially since dysfunctional hearts typically express greater content of N2BA (e.g., diabetes induces an increase in N2BA element content) and thus have greater PEVK content?”*

Response: This is a good point, and we added a paragraph in the Discussion, addressing the above comment (page 17-18, line 543-549): *“When skeletal muscle fibers were stored in glycerol for 4-8 hours, no changes in passive stress were detected. However, a statistically significant increase was found in passive stress after as little as 24 hours of glycerol storage (Figure 1C-E), ranging from 22 % to 124 % depending on sarcomere length, and the observed increase in stiffness was maintained over the full 12 week testing period (Figure 1C-E). Similar findings were made in cardiac muscle. Hence, we recommend that it is best to avoid glycerol for permeabilization and storage.”*

Comment: *“1d. Importantly, the discussion lacks guidance on whether prior work can generally be trusted or if studies mentioning glycerol in their methods need to be replicated using glycerol-free solutions specifically. What model systems might be more reliable than others?”*

Response: Although we only recently investigated in detail the effect of glycerol storage on passive stress (as described in our manuscript under review), we had long suspected its impact based on incidental observations made many years ago. As a result, we deliberately avoided glycerol storage in most of our studies published for the last few decades. Our study under review confirms that this issue primarily affects studies focused on passive stress. Rather than compiling an exhaustive list of studies that have used glycerol storage, we emphasize that this method should be avoided, as it likely leads to inflated passive stress values.

Importantly, we also provide a recommendation for preserving muscle tissues while maintaining their in vivo passive stress characteristics. Several recent studies have demonstrated that rapid freezing in liquid nitrogen, followed by storage at or below -80°C and subsequent thawing in skinning solution, effectively preserves the structure and function of cardiac muscle (Milburn et al., 2022; Ma et al., 2023). However, these studies have primarily focused on active tension measurements. To investigate passive properties,

we conducted a separate set of experiments on muscles that had been rapidly frozen in liquid nitrogen and stored long-term (~5 years) at -80°C.

Upon thawing in skinning solution, one portion of the muscle was maintained in relaxing solution, while the other was stored in glycerol for 3-5 days (further details are provided in the revised manuscript). As shown in Figure 2 below, passive stress in fibers frozen/thawed and then stored in relaxing solution (green triangles) was indistinguishable from that of fresh fibers (i.e., never frozen) and stored in the relaxing solution (blue circles; adapted from Fig 1 in the manuscript). Furthermore, fibers that had been previously frozen and thawed exhibited increased passive stress after 3–5 days of glycerol storage (yellow triangles) — similar to fibers that had not undergone freezing but were stored in glycerol (red circles; adapted from Fig 1 in the manuscript). These findings suggest that rapid freezing followed by long-term storage at -80°C is a viable method for preserving muscle samples while maintaining their passive stress characteristics. In the revised manuscript, we added Methods (page 6, line 161-166), Results (page 13-14, line 411-431, Figure 8), and Discussion (page 18-19, line 598-604).

Figure 2. Changes in passive stress of fresh and frozen muscle that were stored in either relaxing solution or glycerol for ~5 days. Blue and red curves are the results of fresh skinned muscle fibers that were stored in relaxing solution and glycerol, respectively, adapted from the Fig 1C-E in the manuscript. Green and brown curves are the results of frozen(5 yrs)/thawed muscle fibers, stored subsequently in either relaxing solution or glycerol, respectively. Note that the frozen fibers stored in relaxing solution (green curve) exhibited passive stress levels that were indistinguishable from those of fresh muscle stored in relaxing solution (blue curve).

Comment: “2. Differences in Storage Conditions. The samples stored in glycerol are stored at -20 degrees C, while samples stored without glycerol are stored at 4 degrees C. The study would be improved with a control for storage temperature (i.e. glycerol at 4 degrees C).”

Response: We conducted a new set of experiments to address this important point. EDL muscles were stored in either relaxing solution or 50% glycerol, with both conditions maintained at 4°C. After 3–5 days of storage, we measured passive and active stresses (n=10 per storage condition). Consistent with the results in our submitted manuscript (i.e., relaxing solution at 4°C vs. glycerol at -20°C), we observed a significant increase in passive stress in glycerol-stored muscle fibers compared to those stored in relaxing solution, particularly at longer sarcomere lengths (>~2.8 µm) (see Figure 3A-C below). However, there was no difference in active stress or the rate constant of force redevelopment, K_{tr} (Figure 3D and E). We

have included these new findings as a supplemental figure in the revised manuscript and made corresponding updates in the Results section (page 9, line 236-239; page 10, line 262-263), and included the figure below as the Supplementary Figure 1.

Figure 3. Significant increases in peak stress (A), elastic stress (B), and viscous stress (C) of fibers stored in glycerol compared to those in relaxing solution while both storage solutions were kept at 4°C. (n=10 per storage condition; Two-way ANOVA with Tukey's post-hoc analysis). Active stress (D) and rate of force re-development (E) did not show a difference (unpaired t-test).

Comment: “3. Changes in Cell Morphology

3a. Slack sarcomere lengths should be reported to better understand whether the glycation is solely influencing the stretch or whether the slack properties may be changed.

Response: No change was observed in slack sarcomere length in muscle fibers before and after incubation with methylglyoxal, which were $2.367 \pm 0.014 \mu\text{m}$ and $2.366 \pm 0.016 \mu\text{m}$, respectively ($p=0.65$; Paired T-test). We added a sentence to the revised manuscript in the Result section (page 12, line 372-374).

Comment: “3b. Supplemental Figure 1 (Fiber CSA), should also be described in the results.”

Response: We moved the supplemental Figure 1 into the Result in the revised manuscript, as Figure 5, and added a description of results under “Effect of glycerol storage on CSA of muscle fibers”.

Comment: “4. Effect on Active Stress. The authors describe no change in active stress or ktr. However, investigators have noted changes in EC₅₀ with the addition of methylglyoxal (PMID 30333300). A consideration of the limitation that other changes in calcium sensitivity (independent of active stress and ktr) would improve the discussion. The authors may wish to include some context relative to whether the change in passive stiffness may contribute to the explanation but should be cautious not to attribute all change to titin without evidence.”

Response: Thank you for this comment. We agree that investigating potential changes in calcium sensitivity due to glycerol storage is important. To address this, we conducted a new set of experiments

measuring force-pCa relationships in fibers stored in either relaxing solution or glycerol. Our findings indicate that glycerol storage did not affect calcium sensitivity at the short sarcomere length (2.6 μm) that was studied. However, at a longer sarcomere length (3.0 μm), there was a trend toward increased calcium sensitivity in glycerol-stored fibers ($p=0.085$). Moreover, when calculating the ΔpCa_{50} (pCa_{50} at SL 3.0 μm minus pCa_{50} at SL 2.6 μm), glycerol-stored fibers exhibited a significantly greater increase in calcium sensitivity (Figure 4 below). This suggests that glycerol storage enhances the sarcomere length dependence of calcium sensitivity.

These results, along with a new figure, have been incorporated into the Results section under “Effects of glycerol storage on calcium sensitivity” (page 10, line 281-306; Figure 4), and Discussion (page 18, line 577-590). Additionally, we have updated the Methods section to include details on this experiment (page 6, line 137-145).

Figure 4. Calcium sensitivity analysis of fibers stored in relaxing and glycerol solutions. (A) Normalized active stress-pCa curves fitted to the data for each condition. The curves for both

relaxing and glycerol storage solutions at a sarcomere length (SL) of 3.0 μm are left-shifted compared to those at SL 2.6 μm , indicating increased calcium sensitivity at the longer sarcomere length. (B) pCa_{50} values for fibers stored in relaxing solution (blue) and glycerol solution (red) at SL 2.6 μm and 3.0 μm . No significant differences were observed between storage conditions at either sarcomere length, though there was a trend toward higher calcium sensitivity in glycerol-stored fibers at SL 3.0 μm . (Two-way ANOVA.) (C) ΔpCa_{50} (difference in pCa_{50} between SL 3.0 μm and pCa_{50} at SL 2.6 μm). Fibers stored in glycerol showed a greater increase in pCa_{50} than those stored in relaxing solution. (Unpaired t-test.)

Comment: “5a. It is unclear why the investigators used multiple T-tests instead of ANOVA for conditions that included sarcomere length and storage conditions. Please clarify this reasoning “

Response: Thank you for catching this. We first performed a two-way ANOVA, to see interaction between the storage solutions and the sarcomere length as well as the main effects, and then further performed a post-hoc analysis, for example multiple t-test with Bonferroni adjustment. In the revised manuscript, we added the details under Statistics as well as in the Figure legends.

Comment: “5b. This reviewer suggests the inclusion of all F- and p-values, perhaps in the supplement. Inclusion would improve this reviewer's understanding of whether main effects or interactions are significant.” **Response:** All detailed statistical results are now included in the supplementary Excel Spreadsheet which shows F- and P- values as well as t ratio and degree of freedom for each figure.

Comment: “5c. Sample size is noted on some figures as 8-10 fibers. The text should clearly indicate whether these are biological or technical replicates.” **Response:** They are biological variables. We clarified this in our revised manuscript (page 5, line 105-106):

Minor Comments:

Comment: 6. Characterization of Viscous Stress

“6a. Was the viscous stress calculated strictly from the 20-second hold? Because the viscous element is clearly not completely removed, does this severely underestimate the viscous stresses? An estimation of the actual steady-state elastic stress via the plateau of a single or double exponential may be of value.”

Response: In our study, viscous stress was determined after the 20-second hold phase by subtracting the obtained stress value from the peak stress. In response to your comment, we further estimated elastic stress by fitting the stress relaxation data using a double exponential decay curve, taking the plateau value of the fit as the elastic stress. For this analysis, we examined 10 fibers per storage solution, extracted the 10 hold phases per fiber, and fitted each hold phase with a double exponential function to determine the steady-state value. We then performed linear regression and paired t-tests to compare the elastic stress measured after the 20-second hold with the estimated values from the fit (Fig. 5A-B below). As expected, the results showed that, regardless of the storage solution, the estimated elastic stress from double exponential curve fitting was lower than the directly measured elastic stress (Fig. 5A-B). However, the absolute difference between the two values was small (0.05–0.1 mN/mm² on average; Fig. 5B). These findings indicate that our method of estimating viscous stress from the 20-second hold does not affect the overall conclusions of our study.

Figure 5. Comparison between the elastic stress determined after the 20 s hold and estimated elastic stress for the muscle fibers stored in relaxing solution (left panels; blue) and glycerol (right panels; red). For this analysis, ten fibers were analyzed per storage solution and all ten hold phases per fibers were cropped to fit with a double exponential curve (i.e., 100 data points for each plot). A) Linear regression results: the slopes for both regressions were lower than 1.0, indicating that the estimated elastic stress was lower than the measured stress. B) Paired T-test further confirmed that the estimated elastic stress was lower than the measured elastic stress. However, the difference in stress was found to be small; for relaxing solution condition (B, left panel), the measured and estimated stresses were $12.41 \pm 1.0 \text{ mN/mm}^2$ and $12.30 \pm 1.0 \text{ mN/mm}^2$, respectively. For the glycerol storage condition (B, right panel), the measured and estimated stresses were $15.24 \pm 1.2 \text{ mN/mm}^2$ and $15.19 \pm 1.24 \text{ mN/mm}^2$, respectively.

Comment: “6b. Discussion, e.g. page 13. The authors indicate that PEVK glycation is likely present and show that stretch-relaxation of a muscle alters the fibers after step-stretches. Would the authors predict

a different result if using the protocol described in Nedrud Biophys J 2011 (PMID 21943419)? “

Response: Similar to the protocol used in our study, the repeated stretch-release protocol from the Nedrud et al. paper would be expected to result in higher elastic stress and greater hysteresis in glycerol-stored fibers. While we considered demonstrating this experimentally, time constraints prevented us from doing so.

Comment: *“7. Methods- Solutions*

The methods describe 1% Triton X-100 and 50% glycerol. For each, do either addition dilute the relaxing solutions? (For example, is 1.01 mL 100% triton added to 100 mL of relaxing solution or is a concentrated relaxing solution stock used? For glycerol storage, is the glycerol considered a component of the solution or a non-mixing additive making a 1:1 mix of glycerol and normal concentration relaxing solution?) “

Response: For the skinning solution containing 1% Triton X-100, we used a twofold concentrated relaxing solution. Specifically, to prepare 100 mL of skinning solution, we mixed 50 mL of 2× concentrated relaxing solution with 1 mL of Triton X-100 and added 49 mL of distilled water to reach a final volume of 100 mL. For the 50% glycerol storage solution, we treated glycerol as a non-mixing additive and prepared a 1:1 mixture of glycerol and normal-concentration relaxing solution. These details have now been included in the revised manuscript (see page 7, line 173-174).

Reviewer #2 (Comments to the Authors):

Comment: *“The study addresses a longstanding issue, whether glycerol storage has unintended effects on the mechanical properties of skeletal (and possibly cardiac) muscle fibers, and the answer is yes. The authors also provide a mechanism and test this. They point out that these phenomenon may occur in metabolic syndromes as well. It is therefore not only methodologically important but may also have clinical implications. The paper is neat and straightforward. It is definitively an excellent contribution to the field. There are a few minor comments.”*

Response: Thank you for reviewing our study and for the positive evaluation.

Comment: *“At the top of page 9, there is a paragraph that seems to contradict the rest of the paper, in particular this sentence: “Thin-filament extracted fibers had significantly increase peak passive stress, elastic stress and viscous stress (Figs. 3B-D).” My guess is that this is just an editing mistake, but the authors should correct this.”*

Response: Thank you for catching this. Yes, this is an editing problem. We updated the text, see page 11, line 333-335.

Comment: *“The second comment is on the statistics of how the authors compare two curves. They divide the curve up in discrete steps and compare the steps using multiple t-tests. The graphs in figure 1c-e are an example. There is probably some binning of the values on the x-axis as well, which will affect the variation. While there clearly is a difference between the curves, no argument there, I am fairly sure that this is not the appropriate way to do it. The research group has used this method for many years, and it is visually clear, but not correct. If you do this many t-test and properly correct for multiple comparisons (which I don't think the authors have done for the passive force curves), the bar for significance is really high, and you lose a lot of statistical power. I don't have the exact answer of how to do it, but for comparing two curves you normally use regression techniques, so all the xy pairs of the curves are properly weighted, the correlation between the points on the curve is taken into account and therefore*

the statistical power increases greatly. It would also give you confidence intervals instead of error bars. It is however a minor issue for this paper and not essential to its conclusions.”

Response: Thanks to your comment we realized that we did not describe our statistical analysis sufficiently well. To compare passive stress, including the results shown in the figure 1C-E as the reviewer pointed out as an example, we first conducted a Two-way ANOVA to examine main effects and interaction between the storage conditions and the sarcomere length, then further conducted a post-hoc analysis. In the revised manuscript, we added more details in the Methods section and made corresponding changes in figure legends when needed. Furthermore, based on one of the comments of reviewer one, we submit our F-, degree of freedom, and p- values in a supplementary table. Please see supplementary Excel spreadsheet.

Reviewer #3 (Comments to the Authors):

Comment: *“The manuscript by Han et al investigates the effect of glycerol storage on the passive mechanical properties of skeletal muscle fibers, emphasizing the role of titin's PEVK segment. It finds that while glycerol storage preserves active force and crossbridge cycling kinetics, it significantly increases passive tension in fibers due to titin-specific alterations. The research connects these findings to potential pathological implications in metabolic disorders, as similar effects are noted with methylglyoxal-induced lysine crosslinking. Storing muscle and muscle-protein samples in glycerol has been a commonly used method since the ground-breaking era of muscle research in the mid-twentieth century. It has been essentially taken for granted, albeit never tested critically, that striated-muscle structure and function are preserved by this method, which has since been widely employed. The current work thus executes a highly timely test of this age-old hypothesis. The methods are robust, employing standard techniques such as mechanical testing of isolated muscle fibers, gelsolin treatment for thin filament removal, and methylglyoxal exposure to mimic glycation-type modifications. Statistical rigor is high, the large sample size is to be commended, and the tests are executed and interpreted according to the state of the art. The results are discussed, albeit with caution, in the context of potential clinical significance. Altogether, this is an important and straight-forward work. “*

Response: Thank you for reviewing our study and your positive evaluation.

Major comments:

Comment: *“1. The authors analyze the passive muscle-mechanics data in terms of three parameters: peak force, "quasi-steady-state" stress ("elastic stress", measured after 20 seconds of force relaxation), and the difference between them ("viscous stress"). While such a partitioning of data allows for comparison, the terminology should be cautiously used, and the analysis should be carefully discussed. (i) First, such a "passive" mechanical system approaches equilibrium (not steady state), unlike active isometric tension. Therefore, the "quasi-equilibrium stress" may be more appropriate. (ii) Second, it should be kept in mind that the viscous response of the muscle fiber occurs throughout the stretch-and-hold measurement process, thus it alters the peak passive force and it would proceed beyond the 20-second (i.e., the equilibrium is far from set). The arbitrarily chosen 20-second examination window allows for precise comparisons only if the rate of the force decay within this time window is constant. This could actually be easily tested and even used for measuring the glycerol-storage effect. Thus, I recommend (iii)*

to show a comparison of the force decay (for relaxing, glycerol and methylglyoxal situations), and to discuss the above considerations. “

Response: Thank you for the comments and suggestions.

(i) The reviewer is correct that the elastic stress defined in our study is overestimated because passive stress is not at equilibrium after a 20 sec hold and might therefore be better described as “quasi-equilibrium stress”, as we now indicate in the revised manuscript, page 8, line 195-196, as well as figure 1 legend.

(ii) We agree that the viscous response of the muscle fiber occurs throughout the stretch-and-hold measurement process, and we now indicate this in the revised manuscript (page 8, line 196-198). We also compared the measured elastic stress (or quasi-equilibrium stress) to the estimated elastic stress defined as the plateau value obtained by fitting a double exponential decay curve to the stress-relaxation data. Regardless of the storage solutions, the elastic stress estimated from double exponential curve fitting is statistically lower than the elastic stress determined after the 20 s hold (Fig 5A above). However, the absolute difference between the estimated and measured elastic stress is small (0.05 – 0.1 mN/mm² on average; Fig 5B above) and does not impact our conclusions.

(iii) Using data from the experiment with methylglyoxal, we quantified and compared the fast and slow rates of force decay between the two storage solutions before and after methylglyoxal treatment. Interestingly, we observed that methylglyoxal treatment affected the fast rate of force decay of the fibers that were stored in the relaxing solution, while no effect was observed in the glycerol-stored fibers (Figure 6 below). No change was observed in the slow rate of force decay for different storage solution with methylglyoxal incubation. We now show this data in the revised manuscript (page 12-13; line 380-384), as the Supplemental Figure 2.

Figure 6. Fast and slow rates of force decay as a function of storage solution conditions before and after methylglyoxal treatment are shown. Significant reduction in the fast rate of force decay was observed in methylglyoxal treated fibers that had been stored in the relaxing solution (A, paired-t test, $p < 0.0001$), while there was no change observed in the glycerol-stored fibers.

Comment: “2. The authors claim that the glycerol-storage effect is more prominent towards longer sarcomere lengths. While the gap between the control and glycerol curves is indeed widening towards longer sarcomere lengths, notably the entire dataset curves upwards. It would be an easy test to divide the two curves by each other. This way the above claim could be experimentally tested. “

Response: The reviewer's observation that the curve for the glycerol storage condition is consistently higher than the control across all sarcomere lengths is correct. At shorter sarcomere lengths, the percentage increase can appear large due to the near-zero values in fibers stored in relaxing solution. However, this difference typically does not reach statistical significance due to the high variability in the data at shorter sarcomere lengths. In the revised manuscript, we have clarified that the effect of glycerol storage tends to become statistically significant at sarcomere lengths exceeding $\sim 2.8 \mu\text{m}$, where PEVK extension dominates. See the revised manuscript on page 9, line 234-235 (Figure 1 legend).

Comment: "3. The glycerol storage effect is slightly lowered after gelsolin treatment (Figure 3). What might be the reason for it? "

Response: This is a good observation, although it is important to point out that the difference did not reach statistical significance. We consider it possible that a small fraction of the glycerol-storage effect is due to titin-thin filament interactions and that thin filament extraction eliminates this effect. However, if this indeed is the case, the glycerol storage effect on titin-thin filament interaction is small and variable (preventing it from reaching significance in our study).

Comment: "4 In a very nice experiment (Figure 4) the authors show that methylglyoxal treatment of the muscle fiber essentially mimics the glycerol storage effect. I agree that it is tempting to claim therefore that glycerol acts in the same way as methylglyoxal, but I find it premature, simply because glycerol is not a chemical cross-linker. Thus, while the target of glycerol and methylglyoxal might be identical, their mechanisms of action may still be different. The authors may wish to discuss this possibility."

Response: We agree with the reviewer's comment and now indicate this in the revised manuscript (page 16, line 526-530).

Minor comments:

Comment: "1. Using abbreviation in the abstract should be avoided if possible (see EDL)."

Response: Thank you. We spelled out EDL in the abstract.

Comment: "2. The "tr" (tension recovery) in Ktr should be in the subscript throughout the manuscript."

Response: Thank you. We made the change throughout the manuscript.

Comment: "3. Page 7, second paragraph, first sentence: "... (left panels) show passive..." (not "shows")."

Response: Thank you. We have made the correction.

Comment: "4. On page 12, in the third paragraph the authors raise the hypothetical possibility that glycerol treatment may induce a compressive effect on the muscle fiber. By what mechanism? Osmosis?"

Response: Osmotic compression in glycerol and incomplete recovery upon washout is one possibility.

Comment: "5. Page 14, last paragraph, first sentence: I recommend complementing as "...storage effectively preserves the active contractile properties.""

Response: We made the recommended change (page 18, line 577).

In **summary**, we sincerely thank the handling Editor and the three reviewers for their valuable comments, which have helped us further strengthen our manuscript. In response to the feedback, we investigated the glycerol-storage effect on cardiac myocytes (Fig. 1), examined the passive stiffness of frozen-thawed fibers (Fig. 2), assessed the impact of temperature during glycerol storage (Fig. 3), studied calcium sensitivity (Fig. 4), determined elastic stress by fitting the hold-phase (Fig. 5), and analyzed the time constants of fast and slow stress relaxation (Fig. 6). Additionally, we have addressed all other comments by making corresponding revisions in the text of the manuscript.

Thank you again for your thoughtful reviews. We hope that our revised manuscript is now ready for acceptance.

Reference.

- Ma, W., K.H. Lee, C.E. Delligatti, M.T. Davis, Y. Zheng, H. Gong, J.A. Kirk, R. Craig, and T. Irving. 2023. The structural and functional integrities of porcine myocardium are mostly preserved by cryopreservation. *J Gen Physiol.* 155.
- Milburn, G.N., F. Moonschi, A.M. White, M. Thompson, K. Thompson, E.J. Birks, and K.S. Campbell. 2022. Prior Freezing Has Minimal Impact on the Contractile Properties of Permeabilized Human Myocardium. *J Am Heart Assoc.* 11:e023010.

April 10, 2025

Dr. Henk L. Granzier
University of Arizona Medical Center
Dept. of Cellular and Molecular Medicine
1656 East Mabel Street, MRB Room 325
Tucson, AZ 85724-5217

Re: 202413729R1

Dear Henk,

I am pleased to let you know that your manuscript, entitled "Glycerol storage increases passive stiffness of skeletal muscle fibers through effects on the extensibility of titin's PEVK segment." is scientifically acceptable for publication in Journal of General Physiology. All three reviewers have praise for how you and your coauthors responded to their original comments. Reviewer 1 identifies one issue that may merit brief consideration, up to you. Formal acceptance will follow when it is modified in accordance with our editorial policies.

Please note items that need attention are listed at the bottom of this email (under 'manuscript formatting checklist') and on the attached marked-up pdf file. Please also be sure to include a letter addressing the reviewers' comments point-by-point (if applicable) and a copy of the text with alterations highlighted (boldfaced or underlined). Your manuscript should be a double-spaced MS Word file and include editable tables, if appropriate.

Lastly, JGP requires a data availability statement for all research article submissions. These statements will be published in the article directly above the Acknowledgments. The statement should address all data underlying the research presented in the manuscript. Please visit the JGP instructions for authors for guidelines and examples of statements at <https://rupress.org/jgp/pages/editorial-policies#data-availability-statement>.

Please submit your final files via this link:
Link Not Available

Thank you for choosing to publish your research in JGP and please feel free to contact me with any questions.

Sincerely,

Olaf
On behalf of Journal of General Physiology

Journal of General Physiology's mission is to publish mechanistic and quantitative molecular and cellular physiology of the highest quality; to provide a best in class author experience; and to nurture future generations of independent researchers.

Manuscript formatting checklist:

- MS Word document of text needed (including editable tables)
 - MS Word document of supplemental text needed, if applicable (including figure legends and editable tables)
 - Brief Statement describing supplementary information needed, if applicable (in subsection at end of Materials & Methods)
 - Please include a data availability statement preceding the Acknowledgments section. Please see <https://rupress.org/jgp/pages/editorial-policies#data-availability-statement>
 - Figures created at sufficient resolution and in acceptable format (including supplemental if applicable). If working in Illustrator, we prefer .ai or .eps file format. If working in Photoshop please use 600dpi/1000dpi .tiff or .psd file format. Minimum resolution at estimated print size: Minimum resolution for all figures is 600 dpi. For figures that contain both photographs and line art or text, 600 dpi is highly recommended. Figures containing only black and white elements (line art, no color, and no gray) should be 1,000 dpi. Maximum figure size is 7 in wide x 9 in high (17.5 x 22.8 cm) at the correct resolution. <https://jgp.rupress.org/fig-vid-guidelines>
 - Supplemental figures, if any, conforming to same guidelines as manuscript figures (noted above)
 - If images resemble one from a prior publications, the author must seek permissions (to reproduce or adapt) from the original publisher. [You can resubmit your paper while waiting to hear back from the original publisher but please keep us updated]
 - All authors must complete a disclosure form prior to acceptance. A link to complete the form has been sent to all coauthors. Please provide the editorial office with updated email addresses if necessary
-

Reviewer #1 (Comments to the Authors):

The study, "Glycerol storage increases passive stiffness of skeletal muscle fibers through effects on the extensibility of titin's PEVK segment" by Han et al has been substantially revised. The manuscript includes further evidence that glycerol storage of permeabilized fibers impacts passive mechanics of striated muscle.

The authors have addressed this reviewer's substantial comments. One follow up comment/clarification:

Comment 6a. Viscous stress. This reviewer accepts that the fit should correlate well with the 20 second hold data. However, if fit using (2-)exponential or power-law models, the viscous stress is unlikely to be fully resolved in 20 seconds. While no new data is likely needed, it is possible that other investigators using shorter or longer hold periods may find slightly different results by under or over estimating the viscous stress.

Reviewer #2 (Comments to the Authors):

The authors responded to all the questions raised in the review. The responses were satisfactory, and I recommend to accept the article for publication

Reviewer #3 (Comments to the Authors):

The authors did an outstanding job in responding to the critical questions and comments. They carried out additional measurements to back up their arguments. The manuscript has increased in quality, and provides a nicely rounded work.

Reviewer #1 (Comments to the Authors):

"The study, "Glycerol storage increases passive stiffness of skeletal muscle fibers through effects on the extensibility of titin's PEVK segment" by Han et al has been substantially revised. The manuscript includes further evidence that glycerol storage of permeabilized fibers impacts passive mechanics of striated muscle.

The authors have addressed this reviewer's substantial comments. One follow up comment/clarification:

Comment 6a. *Viscous stress. This reviewer accepts that the fit should correlate well with the 20 second hold data. However, if fit using (2-)exponential or power-law models, the viscous stress is unlikely to be fully resolved in 20 seconds. While no new data is likely needed, it is possible that other investigators using shorter or longer hold periods may find slightly different results by under or over estimating the viscous stress."*

Response: We thank the Reviewer 1 for the follow-up comment. We agree with the reviewer's view.

Reviewer #2 (Comments to the Authors):

"The authors responded to all the questions raised in the review. The responses were satisfactory, and I recommend to accept the article for publication."

Response: We thank Reviewer 2 for recommending acceptance of our work for publication.

Reviewer #3 (Comments to the Authors):

"The authors did an outstanding job in responding to the critical questions and comments. They carried out additional measurements to back up their arguments. The manuscript has increased in quality, and provides a nicely rounded work."

Response: We thank the Reviewer 3 for the positive evaluation of our revised manuscript.